# Pupil-linked arousal is driven by decision uncertainty and alters serial choice bias

Anne E. Urai[1,2], Anke Braun[1] & Tobias H. Donner[1,2,3]

While judging their sensory environments, decision-makers seem to use the uncertainty about their choices to guide adjustments of their subsequent behaviour. One possible source of these behavioural adjustments is arousal: decision uncertainty might drive the brain's arousal systems, which control global brain state and might thereby shape subsequent decision-making. Here, we measure pupil diameter, a proxy for central arousal state, in human observers performing a perceptual choice task of varying difficulty. Pupil dilation, after choice but before external feedback, reflects three hallmark signatures of decision uncertainty derived from a computational model. This increase in pupil-linked arousal boosts observers' tendency to alternate their choice on the subsequent trial. We conclude that decision uncertainty drives rapid changes in pupil-linked arousal state, which shape the serial correlation structure of ongoing choice behaviour.

[1] Department of Neurophysiology and Pathophysiology, University Medical Center Hamburg-Eppendorf, Hamburg 20246, Germany. [2] Department of Psychology, University of Amsterdam, Amsterdam 1018 WT, The Netherlands. [3] Amsterdam Brain and Cognition (ABC), University of Amsterdam, Amsterdam 1018 WT, The Netherlands. Correspondence and requests for materials should be addressed to A.E.U. (email: anne.urai@gmail.com) or to T.H.D. (email: t.donner@uke.de).

In perceptual and sensory-motor tasks, humans and animals behave as if they make use of decision uncertainty—the probability that a choice is correct, given the sensory evidence[1–3]. Theoretical accounts postulate that decision uncertainty should shape subsequent decision processing and, thereby, subsequent choice behaviour[1,4,5]. But how decision uncertainty is transformed into subsequent behavioural adjustments has, so far, remained elusive.

One prominent idea is that the brain broadcasts uncertainty signals across brain-wide neural circuits via low-level arousal systems[4,6,7]. Arousal systems might be driven by uncertainty[4,7–11], and they profoundly shape the global state of the brain through the action of modulatory neurotransmitters[12–14]. Uncertainty-dependent changes in global brain state, in turn, might translate into adjustments of choice behaviour. The goal of our study was to investigate whether arousal (1) reflects decision uncertainty in a perceptual choice task; and (2) predicts changes in subsequent choice behaviour.

Changes in central arousal state (as assessed by various measures of cortical dynamics) are tightly coupled to fluctuations in pupil diameter under constant luminance[13,15–18]. We here built on this connection and monitored pupil diameter as a proxy for central arousal state. We used a model based on statistical decision theory, illustrated in Fig. 1, in which decision uncertainty is defined as the probability a choice is correct, given the available evidence[1,19]. This operationalization of decision uncertainty obviates the need for subjective confidence reports[5], bridging to the insight from animal physiology that neurons in a number of brain regions encode decision uncertainty, as defined in Fig. 1 (refs 2,20–22).

The model assumes that observers base their judgment of each stimulus on a noisy decision variable, sampled from a distribution that depends on the identity and strength of the stimulus (Fig. 1a). Two-alternative forced choice tasks entail comparing this decision variable with a decision bound. When the decision variable happens to fall on the wrong side of the bound, errors occur. This happens more often for weaker stimuli, because the distributions corresponding to the two possible stimuli show higher overlap (Fig. 1b). A monotonic function of the distance between the decision variable and the bound is a metric of decision confidence; uncertainty is its complement[2,19,23] (Fig. 1a and Methods).

This model predicts three signatures of decision uncertainty[2,19]: (1) uncertainty decreases with evidence strength for correct choices (blue line in Fig. 1c) but, counter-intuitively, increases with evidence strength for incorrect choices (red line in Fig. 1c); (2) uncertainty predicts a monotonic decrease in choice accuracy from 100 to 50% (Fig. 1d); (3) higher uncertainty predicts lower choice accuracy, even for the same evidence strength (Fig. 1e). The opposite, monotonic scaling of uncertainty with evidence strength for correct and error trials (Fig. 1c) also emerges from a variety of dynamic decision-making models, including race models[2], Bayesian attractor models[24], and biophysically detailed circuit models of cortical dynamics[25,26].

We systematically manipulated the strength of sensory evidence and tested whether pupil responses exhibited the three signatures derived above. We then quantified the predictive effects of pupil-linked arousal on subsequent behaviour in terms of the key elements of the perceptual decision process: response time (RT), perceptual sensitivity, lapse rate, and choice bias. Choice bias was decomposed into an overall bias for one choice, and a serial bias dependent on the history of previous choices or stimuli. We found a predictive effect of pupil-linked arousal responses on serial choice bias.

## Results

**Pupil responses reflect decision uncertainty.** Twenty-seven human observers performed a two-interval forced choice visual motion coherence discrimination task (Fig. 2a and Methods). We applied motion energy filtering[27] to the stochastic random dot motion stimuli, yielding a more fine-grained estimate of the decision-relevant sensory evidence contained in the stochastic stimuli than the nominal level of motion coherence (Fig. 2b,c and Methods). The absolute value of this sensory evidence served as a single-trial measure of evidence strength (Fig. 2b). As expected, stronger evidence yielded higher choice accuracy and faster responses (Fig. 2d and Supplementary Fig. 2a).

In line with previous work[19], RT exhibited all three signatures of decision uncertainty derived in Fig. 1 above (Fig. 2e and

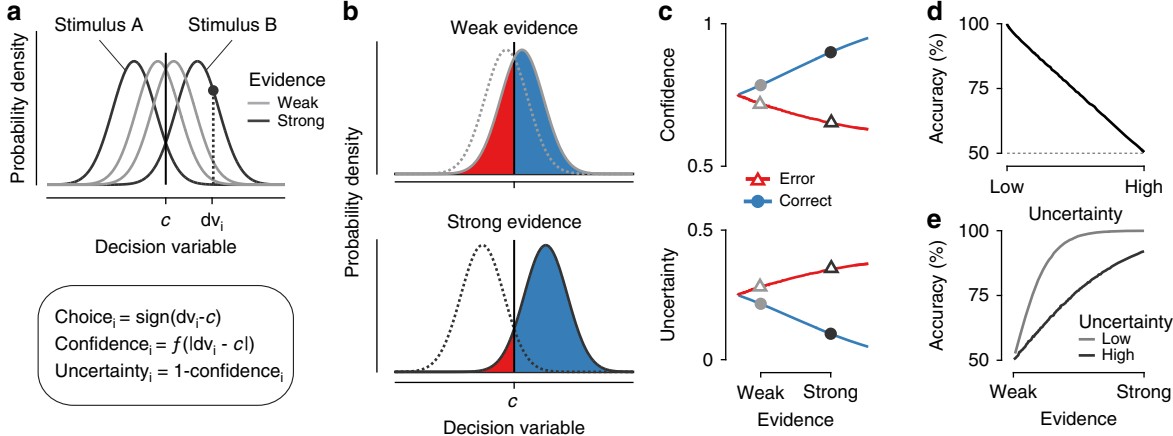

**Figure 1 | Operationalizing decision uncertainty.** (**a**) Computations underlying choice and decision uncertainty. Due to noise, repeated presentations of a generative stimulus produce a normal distribution of internal responses centred at the mean of this generative stimulus. The internal response on each trial $dv_i$ is a sample drawn from this distribution. It is compared with a decision bound or criterion $c$, to compute the binary choice as well as a graded measure of decision confidence (or its complement: uncertainty). (**b**) For two example levels of evidence strength, the average confidence is indicated by the shaded regions, separately for correct (blue) and error (red) trials. (**c**) Confidence (top) and uncertainty (bottom) as a function of evidence strength (100 bins), separately for correct and error trials. The two levels of evidence indicated by symbols (circles, triangles) correspond to the two example levels of evidence strength in **a**,**b**. (**d**) Accuracy as a function of decision uncertainty (100 bins). (**e**) Accuracy as a function of evidence strength (100 bins), separately for trials with high and low decision uncertainty (median split). For details, see Methods and refs 2,19.

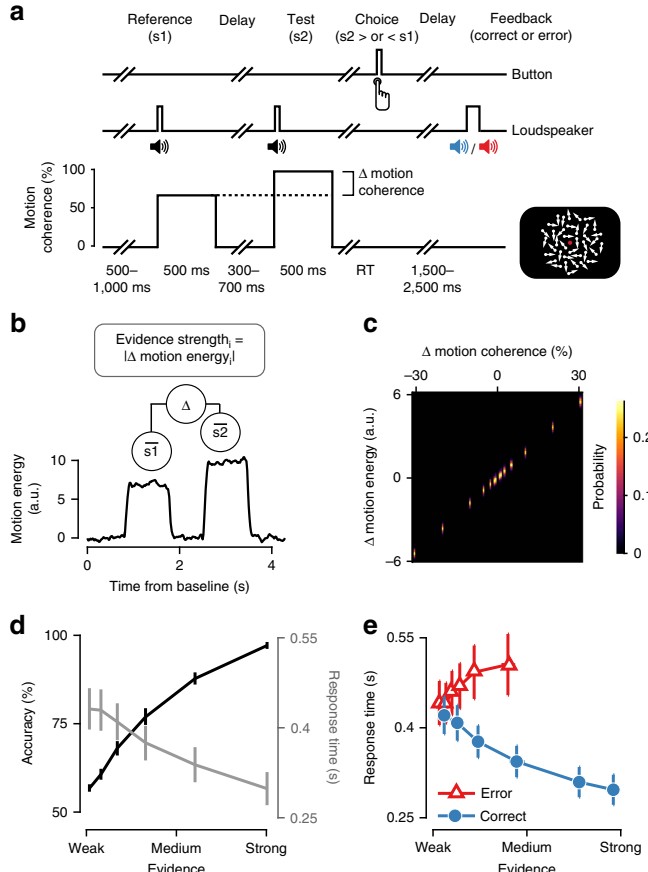

**Figure 2 | Perceptual choice task and behaviour. (a)** Behavioural task. Dynamic random dot patterns were displayed throughout each trial. In two successive intervals (onset cued by beeps), the dots moved in one of the four diagonal directions (fixed per observer): A first 'reference' interval with always 70% motion coherence, and a second 'test' interval with varying levels of motion coherence, larger or smaller than the reference. Observers reported whether the test stimulus contained stronger or weaker motion than the reference by pressing one of two buttons. They received auditory feedback after a variable delay. **(b)** Quantifying evidence strength. Each random dot stimulus was convolved with a set of spatio-temporal filters[27] to obtain a time course of motion energy. The difference between mean motion energy during test and reference intervals was used as a measure of single-trial measure evidence strength. **(c)** Probability distribution of evidence strength as a function of difference in nominal motion coherence. **(d)** Accuracy and median RT as a function of evidence strength (six bins). **(e)** Median RT as function of evidence strength (six bins), split by correct and error trials. ($N = 27$, group mean ± s.e.m.).

Supplementary Fig. 1b,c). This was true despite the interrogation protocol[28], in which the test stimulus had a fixed duration, its offset prompted the choice, and observers were instructed to maximize accuracy without speed pressure (response deadline was 3 s after test offset). Specifically, RT decreased with evidence strength on correct trials but increased with evidence strength on errors (Fig. 2e). Further, RT predicted accuracy over a wide range, but not below 50% (Supplementary Fig. 1b), indicating that RT reflected decision uncertainty rather than error detection[2]. We next assessed whether decision uncertainty also affected pupil-linked arousal.

The pupil dilated during decision formation, peaking just after the choice (button press) as observed in previous work[29], and then dilated again after feedback (Fig. 3a). Between these two peaks, dilation amplitudes diverged between different conditions,

as predicted by decision uncertainty (compare with Fig. 1c): Pupil responses were smallest after correct decisions based on strong evidence, they were overall larger after errors than correct choices, and largest after errors made on trials with strong evidence (Fig. 3a).

To quantify the temporal evolution of uncertainty scaling in the pupil, we regressed baseline-corrected pupil time courses against each trial's evidence strength, separately for correct and error trials. From choice onwards, pupil dilation scaled positively with evidence strength on error trials, and negatively on correct trials (Fig. 3b,c and Supplementary Fig. 3a). In other words, the scaling of the pupil response with evidence strength diagnostic of decision uncertainty emerged in the interval between choice and feedback. Consequently, this uncertainty scaling was not a response to the external information about choice correctness provided by the external feedback, but rather reflected internal decision-related computations as described in Fig. 1. For simplicity, we refer to the single-trial pupil dilation averaged across the 250 ms interval before feedback as 'pupil response' in the following.

The pupil response also exhibited the other two signatures of decision uncertainty predicted by the model in Fig. 1. Larger pupil responses were accompanied by an overall lower choice accuracy (Fig. 3e and Supplementary Fig. 3c), and psychophysical sensitivity was lower on trials with a larger pupil response (Fig. 3d and Supplementary Fig. 3b). Specifically, the pupil response did not predict choice accuracy below 50%, suggesting that it did not signal the detection of errors (Supplementary Fig. 3c).

The scaling of the pupil response with decision uncertainty was not inherited from the analogous scaling of RT, but was also present after first removing (via linear regression) the trial-to-trial variations accounted for by RT (Supplementary Fig. 3d–f). Indeed, trial-to-trial correlations between pupil responses and RTs were generally small (Pearson correlation, average $r$: 0.087 range: $-0.042$ to 0.302, for log-transformed RT). For all subsequent analyses reported in this paper, we removed RT-fluctuations from the trial-to-trial fluctuations of single-trial pupil responses via linear regression (see Methods).

**Pupil-linked arousal alters subsequent choice behaviour.** We proceeded to test whether uncertainty-related pupil responses predicted changes in subsequent choice behaviour. It has been proposed that arousal signals control various aspects of learning and decision-making[4,6–8,11]. In the context of our task, the choice parameters of interest were perceptual sensitivity (measured as the slope of the psychometric function, Supplementary Fig. 4a), lapse rate (measured as the vertical distance of the asymptotes of the psychometric function from 0 or 1, Supplementary Fig. 4a), bias (measured as the horizontal shift of the psychometric function, Supplementary Fig. 4a) and RT. For RT, we focussed on increases after error trials, an effect referred to as post-error slowing[30], which was found to be modulated by pupil-linked arousal in a speeded RT task[31]. Choice bias was further decomposed into two parameters: overall bias (that is, a general tendency towards one choice option, averaged across the entire experiment, Supplementary Fig. 4b) and serial bias (that is, a local, choice history-dependent tendency towards one option that becomes evident when conditioning the psychometric function on the preceding choice, Supplementary Fig. 4c (refs 32–34)). Because in our task (as common in laboratory choice tasks), the sensory evidence was independent across trials, any serial bias was maladaptive, reducing observers' performance below the optimum they could achieve given their perceptual sensitivity.

The pupil response predicted a reduction of serial choice bias (Fig. 4a and Supplementary Fig. 5). When a choice was followed

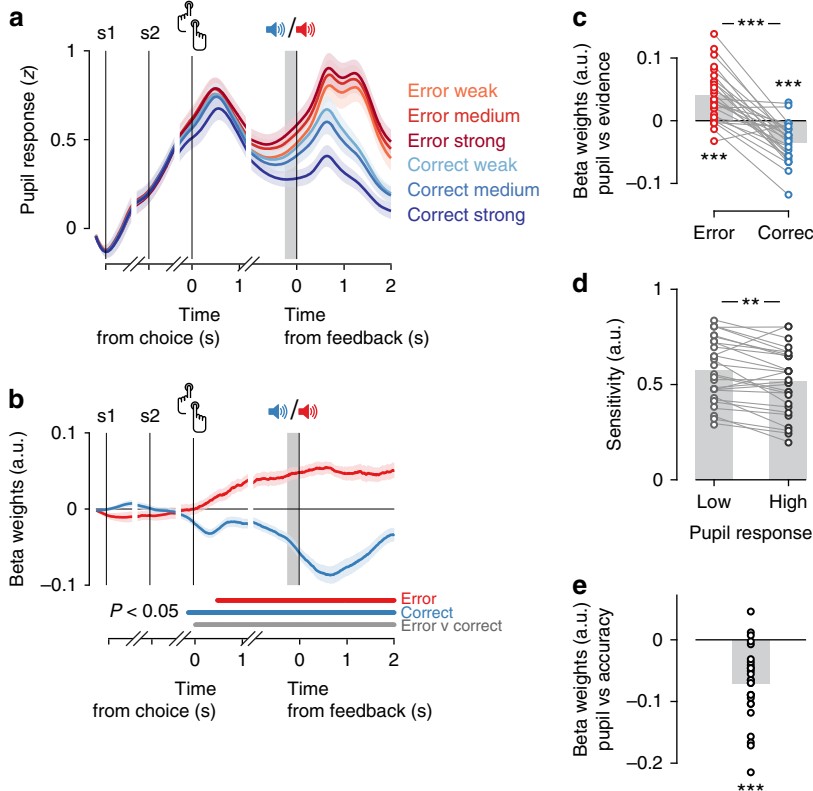

**Figure 3 | Pupil dilation after choice and before feedback reflects decision uncertainty.** (**a**) Time course of pupil responses throughout the trial. Time courses were baseline-corrected and split by correct and error as well as three bins of evidence strength. Mean pupil dilation in the 250 ms before feedback (grey box) was used as a single-trial measure of pupil response. (**b**) Time course of uncertainty scaling in the pupil, computed as sample-by-sample regression of baseline-corrected pupil dilation onto evidence strength. Lower bars indicate $P < 0.05$ from a cluster-corrected permutation test, of the difference between each time course and zero, and between the two time courses. (**c**) Regression weights for the linear relationship between evidence strength and pupil responses. (**d**) Individual perceptual sensitivity, separately for lowest and highest pupil tertiles. (**e**) Individual logistic regression weights, using pupil responses to predict single-trial choice correctness. In **b–e** $z$-scored, log-transformed RTs were removed from the pupil signal via linear regression. ***$P < 0.001$, **$P < 0.01$, permutation test. ($N = 27$, group mean ± s.e.m.).

by a small pupil response, observers tended to repeat this choice on the next trial; when the previous pupil response was large, this serial bias was abolished (Fig. 4a). This predictive effect was similar for correct and error trials (Supplementary Fig. 6a). An analogous pattern of predictive effects was observed when binning by previous trial RT: Fast, but not slow, RTs were followed by a tendency to repeat the previous choice (Fig. 4f and Supplementary Fig. 6b).

The pupil response predicted none of the other choice parameters on the next trial (assessed by one-way repeated-measures analysis of variance (ANOVA)), neither overall choice bias (signed overall bias: $F_{(2,52)} = 0.939$, $P = 0.398$, $Bf_{10} = 0.221$; absolute value of overall bias: $F_{(2,52)} = 1.817$, $P = 0.173$, Fig. 4b), nor perceptual sensitivity ($F_{(2,52)} = 1.936$, $P = 0.155$, Fig. 4c), nor lapse rate ($F_{(2,52)} = 2.213$, $P = 0.120$, Fig. 4d), nor RT (overall RT: $F_{(2,52)} = 3.232$, $P = 0.048$, $Bf_{10} = 1.207$; post-error slowing: $F_{(2,52)} = 2.056$, $P = 0.138$, Fig. 4e). Variations in RT, likewise, did not predict a change in any of the other parameters of the decision process (Fig. 4g–j, all $P > 0.05$). The overall pattern of results implies that observers did not simply act more randomly after large pupil responses or RT. Random button presses would have reduced sensitivity, in other words, decreased the slope of the psychometric function, contrary to our observations (Fig. 4c,h). Rather, the pattern of results implies that, after large pupil responses or RT, observers' tendency towards one or the other choice became less history-dependent.

In sum, large pupil responses and slow RTs were neither followed by improved processing of sensory evidence (a common effect of attention[35]), nor a change in overall response bias. Large pupil responses and slow RTs were followed by only minor (and statistically not significant) changes in stimulus-independent lapses as well as small adjustments in speed-accuracy trade-off, as observed after response conflict, errors, or large pupil responses in speeded RT tasks[31,36,37]. The weak effect on post-error slowing might be due to the use of an interrogation protocol in our study, which did not require observers to optimize their speed-accuracy trade-off[28]. However, both RT and pupil-linked arousal had a robust effect on serial choice bias, reducing an overall repetition bias that predominated across the group of observers. This effect of both uncertainty-related measures on the serial correlation structure of choice behaviour has so far been unknown. We therefore proceeded to model and comprehensively quantify this effect at the level of individual observers.

**Pupil-linked arousal predicts choice alternation.** To this end, we extended a previously established regression model of serial choice biases[33] with pupil- and RT-dependent modulatory effects. The basic model (that is, without modulatory terms) quantified the impact of the previous seven choices and stimuli on the current choice bias in terms of linear combination weights (Fig. 5a, see

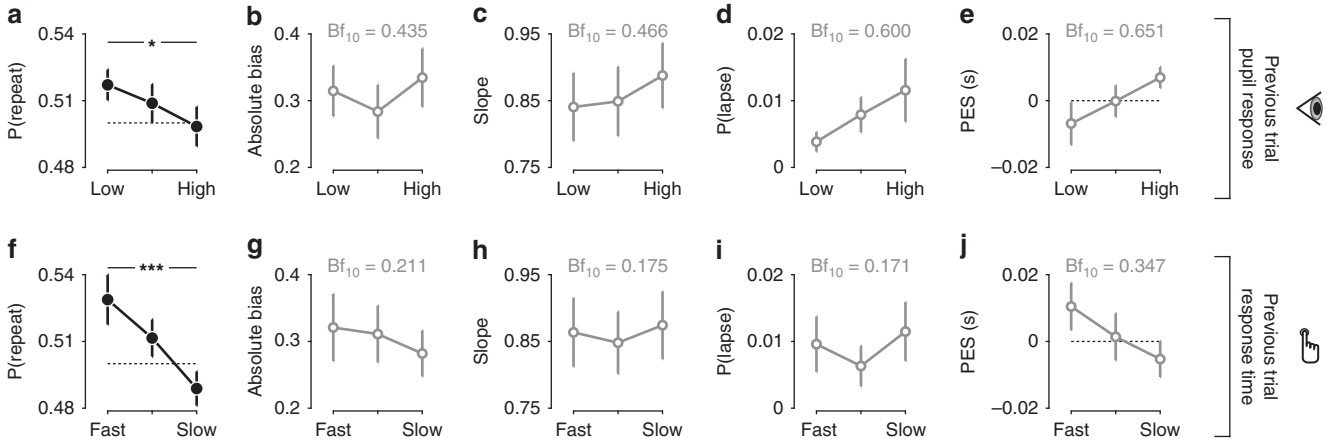

**Figure 4 | Pupil responses and RT predict reductions in serial choice bias.** (**a**) Serial choice bias, quantified as the history-dependent shift of the psychometric function, for tertiles of previous trial pupil responses. (**b**) Absolute choice bias, measured as the intercept of a logistic psychometric function, for tertiles of previous trial pupil responses. (**c**) Perceptual sensitivity, measured as the slope of a logistic psychometric function, for tertiles of previous trial pupil responses. (**d**) Lapse rate, measured as the probability of stimulus-independent guesses, for tertiles of previous trial pupil responses. (**e**) Post-error slowing, for tertiles of previous trial pupil responses. (**f–j**) as in **a–e**, but for tertiles of previous trial RT. ***$P < 0.001$, *$P < 0.05$, main effect of pupil/RT bin on repetition probability computed from a one-way repeated measures ANOVA. Unfilled markers indicate $P > 0.05$, with $Bf_{10}$ from a Bayesian repeated measures ANOVA written in panel. ($N = 27$, group mean ± s.e.m.).

Methods and ref. 33). We added to this model multiplicative interaction terms, that quantified how much the effect of previous stimuli and choices was modulated by either pupil response or RT on those same trials (Fig. 5a). Simultaneously modelling the effects of both pupil responses and RT enabled us to estimate their independent impact on serial choice bias; we found the same results when fitting a separate regression model for each modulatory variable (Supplementary Fig. 7).

The model fits revealed robust, and idiosyncratic, patterns of serial choice biases in most observers (Fig. 5c,d; see Supplementary Fig. 2b,c for individual sessions). As expected, the contribution of past stimuli and choices to current behaviour was strongest when sensory evidence was weak and decayed strongly with evidence strength (Fig. 5b). The weight of the immediately preceding choice was generally stronger than the weight of the previous stimulus (Fig. 5d). The effect of previous choices lasted up to seven trials in the past (corresponding to about 60 s, Fig. 5c), but had the largest absolute magnitude on the preceding trial (Fig. 5c, grey dashed line). There was large inter-individual variability in choice weights (Fig. 5c,d). While the majority of observers systematically repeated their choices (purple symbols; 12 significant at $P < 0.05$), a good fraction tended to alternate their choices (orange symbols; 7 significant at $P < 0.05$).

Observers' serial choices biases were unrelated to the (small) serial correlations between stimuli. The transition probabilities between stimulus categories (that is, s2 > s1 or s2 > s1) were close to 0.5 (range across observers: 0.475–0.508), and did not correlate with individual choice weights (Pearson correlation $r = 0.010$, $P = 0.960$, $Bf_{10} = 0.149$) or stimulus weights (Pearson correlation $r = -0.176$, $P = 0.381$, $Bf_{10} = 0.217$). Likewise, the auto-correlation of absolute motion coherence differences (that is, absolute levels of evidence strength) was close to 0 (range across observers: $-0.061$ to 0.028) and did not correlate with individual choice weights (Pearson correlation $r = 0.123$, $P = 0.541$, $Bf_{10} = 0.179$) or stimulus weights (Pearson correlation $r = -0.142$, $P = 0.480$, $Bf_{10} = 0.190$).

Critically, pupil responses and RT both negatively interacted with the effect of previous choices (Fig. 5e), in line with the observation that large pupil responses or long RTs were followed by less choice repetition (Fig. 4a,f). By contrast, neither pupil

responses nor RT interacted with the effect of the previous stimulus (Fig. 5e). Pupil responses beyond one trial in the past, as well as baseline pupil diameter on the current trial, did not predict a modulation of serial biases (Supplementary Fig. 8). Moreover, these results were not accounted for by trial-to-trial variations in trial timing or the passage of time between trials (Supplementary Fig. 9).

The pupil response after feedback did not contain information predictive of serial choice bias, beyond the information already present during the pre-feedback interval. The post-feedback pupil responses similarly predicted modulation of serial choice biases, but no longer did so when removing (via linear regression) variance explained by pre-feedback pupil responses from the post-feedback pupil signal (Supplementary Fig. 10).

While the modulatory effects associated with pupil responses and RT were both negative on average, such an overall reduction of the group-level repetition bias (Fig. 4a,f) might be due to two alternative scenarios at the level of individual observers: either a reduction of each observer's intrinsic serial choice bias for repetition or alternation (referred to as 'bias reduction' hereafter); or, alternatively, a general boost of choice alternation, regardless of the observer's intrinsic serial bias (referred to as 'alternation boost'). We quantified intrinsic serial bias as each observer's choice weight (that is, the main effect of the previous on the current choice estimated by our model). The bias reduction scenario predicts a negative correlation between choice weights and modulation weights across observers. The alternation boost scenario predicts negative individual modulation weights for all observers, independently of their corresponding choice weights (that is, no correlation).

The analysis of these individual behavioural patterns revealed dissociable effects of pupil-linked arousal and RT (Fig. 5f,g). Modulation weights for the pupil were negative for most observers, irrespective of their individual choice weight. When splitting all 27 observers into 'alternators' and 'repeaters' based on the sign of their intrinsic bias (that is, choice weight), we found no correlation between individual modulation and choice weights (Fig. 5f, Pearson correlation $r = 0.017$, $P = 0.935$, $Bf_{10} = 0.149$). Further, the modulation weights were negative for both subgroups, and not significantly different between them

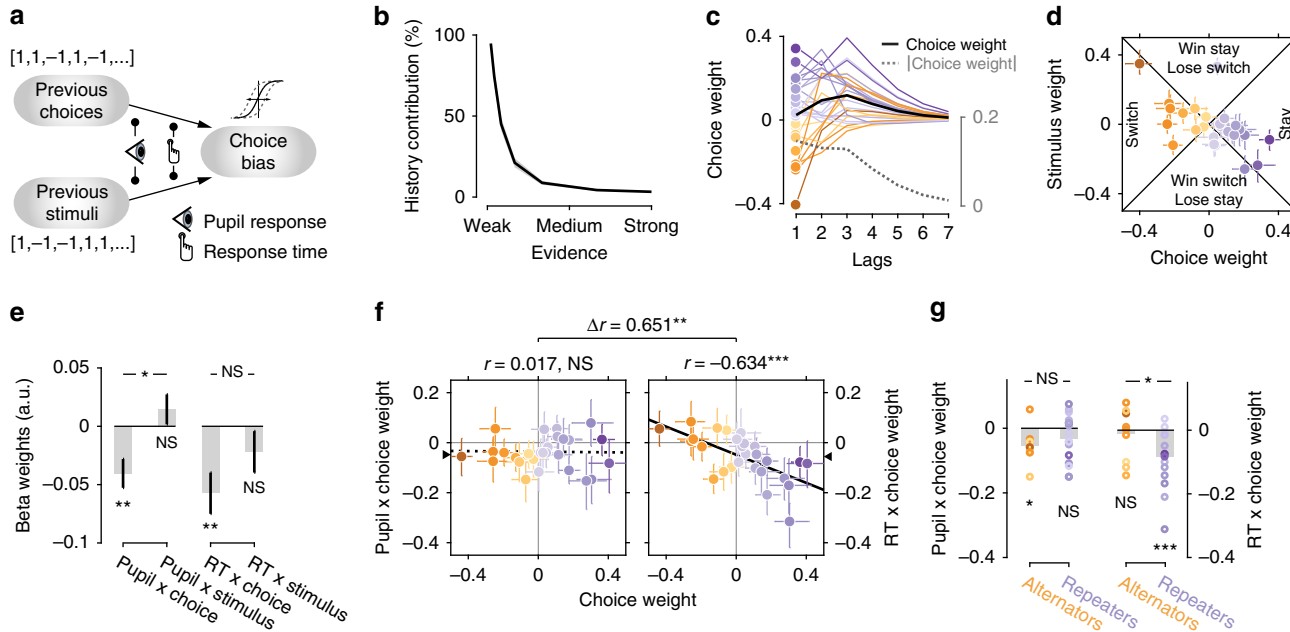

**Figure 5 | Modelling the modulation of serial choice bias.** (**a**) Schematic representation of the regression model with modulatory terms. (**b**) The contribution of history terms (past choices and stimuli) as a fraction of the total variance in the decision variable[33], decreased with stronger sensory evidence. (**c**) Choice weights for the previous seven trials, obtained from the history model without modulatory terms. Each line corresponds to one observer. Purple, 'repeaters' with positive choice weight for lag 1. Orange, 'alternators' with negative choice weights for lag 1. Black line, group mean. Grey dashed line, group mean of absolute choice weight. (**d**) Choice weights at lag 1 plotted against the corresponding stimulus weights. Coloured dots and error bars indicate individual observers ± 68% confidence intervals obtained from a bootstrap. See Methods for an interpretation of this graph in terms of behavioural strategy. (**e**) Regression weights for the interaction between previous pupil response or RT and previous choices or stimuli. N = 27, group mean ± s.e.m. (**f**) Correlation between choice weights and their modulation by pupil dilation or RT. Colours indicate the choice weight as derived from the basic model in **d**. Error bars are 68% confidence intervals obtained from a bootstrap. The intercept of the least-squares regression line, corresponding to the mean beta weight across the group, is indicated with a triangle on the y axis. (**g**) Beta weights for interaction between previous pupil response or RT and previous choices. Group split based on the sign of individual choice weights. ***$P < 0.001$, **$P < 0.01$, *$P < 0.05$, NS $P > 0.05$, Pearson's correlation coefficient or permutation test.

(Fig. 5g). These observations are consistent with the idea that pupil-linked arousal generally boosted observers' tendency to alternate their choice on the next trial.

The alternation boost scenario for pupil responses was further supported by a striking contrast to RT-linked modulations, which were in line with the bias reduction scenario. The RT-linked modulation weights exhibited a strong negative correlation with individual choice weights (Fig. 5f, Pearson correlation $r = -0.634$, $P < 0.001$, $Bf_{10} = 76.359$), were negative only for the group of repeaters, and differed significantly between alternators and repeaters (Fig. 5g). Correspondingly, the correlations with individual choice weights were significantly different for pupil- and RT-modulation weights (Fig. 5f). Moreover, RT-dependent bias reduction was most pronounced after incorrect choices, whereas the pupil-dependent alternation boost was most pronounced after correct choices (Supplementary Fig. 11).

In sum, the modulatory effects associated with post-decision pupil-linked arousal and RT both shaped the serial correlation structure of choices, but in distinct ways: pupil-linked arousal generally promoted choice alternation, regardless of the observer's intrinsic bias, whereas RT-linked processes generally reduced observers' intrinsic bias.

## Discussion

Decisions about an observer's sensory environment do not only depend on the momentary sensory input but also on the behavioural context[38]. One such contextual factor is the history of preceding choices and stimuli, which robustly biases even

highly trained decision-makers[33]. Although such serial choice biases were first identified in psychophysical tasks about a century ago[32], their determinants have remained poorly understood. Previous treatments of serial choice biases have conceptualized experimental history as sequences of binary external events (past stimulus identities, choices, or feedback)[33,39]. We here established that these serial biases were also modulated by the decision-maker's pupil-linked arousal state on the previous trial, which, in turn, reflected the uncertainty about the observer's choice.

Several important features of our approach allowed us to move beyond previous work linking human pupil dynamics to uncertainty and performance monitoring. First, different from most previous studies, we here unravelled the temporal evolution of uncertainty information in the pupil response, enabling inferences about not only the existence, but also the time course of this information (see ref. 40 for a similar approach). Second, the model-based definition of decision uncertainty we used helped dissociate decision uncertainty from error detection, which has previously been linked to pupil dilation[41]. In a two-choice task, a signal encoding decision uncertainty should predict behavioural performance over a range from 100% to 50% correct (corresponding to 50% for the maximum uncertainty signal, or larger when encoding is imprecise). By contrast, an error detection signal should predict performance over the range 100% to 0% correct[2]. Our measurements were more consistent with decision uncertainty than error detection (Supplementary Fig. 3c). Third, in our task, decision uncertainty critically depended on internal noise (the primary source of the variance in Fig. 1a). By contrast, previous studies linking uncertainty to pupil dynamics[9,10,40,42]

have used tasks in which the primary source of uncertainty was in the observers' environment. Last, in contrast to most previous pupillometry studies[29,42,43] we comprehensively quantified the predictive effects of pupil-linked arousal on the parameters of choice beyond the current trial, thereby complementing recent work on the effects of pupil-linked arousal on learning[9,40]. Taken together, our results critically advance the understanding of how internal decision uncertainty is encoded in pupil-linked arousal in humans, in a way that builds a direct bridge to single-unit recording studies of decision uncertainty in animals[2,20–22].

The neural sources of task-evoked pupil responses at constant luminance are not yet fully identified[44], but mounting evidence points to the noradrenergic locus coeruleus (LC)[45–47] (a core component of the brain's arousal system[11]) as well as the superior and inferior colliculi[48]. Microstimulation of all three structures triggers pupil dilation[45]. Among these structures, activity of the LC (spontaneous or evoked by electrical stimulation) is followed by pupil dilation at the shortest latency[45]. The LC also has widespread, modulatory projections to the cortex implicated in regulating central arousal[11]. Dopaminergic and cholinergic systems, which are closely connected with the LC[49], are likewise implicated in central arousal state[13] and may also contribute to task-evoked pupil responses.

We propose that decision-makers' uncertainty about their choices might shape serial choice biases by recruiting pupil-linked neuromodulatory systems. Frontal brain regions encoding decision uncertainty send descending projections to several of these systems[11,49], which in turn project to large parts of the cortex, including networks of regions involved in perceptual inference and decision-making[50]. Neuromodulators like noradrenaline can profoundly alter the dynamics and topology of cortical networks[13,15,51,52]. Thus, these brainstem arousal systems might be in an ideal position to transform variations in decision uncertainty into adjustments of choice behaviour[4,7].

The behavioural effect of pupil-linked arousal might be explained by at least two (not mutually exclusive) scenarios. First, arousal responses might promote choice alternation at the level of response preparation, by altering the state of the motor system[53]. Second, the arousal response might modulate the decision stage—specifically the dynamic updating of beliefs about the upcoming evidence, for example by shifting the criterion (assumed to be constant in signal detection theory, Fig. 1) from one choice to the next. When this criterion is shifted in the direction opposite to the last choice, alternation ensues. In line with these ideas, changes in pupil-linked arousal state can indeed translate into specific behavioural effects[15,29], presumably by interacting with selective cortical circuitry[54].

Our current observations are not easily reconciled with existing theoretical accounts of the impact of phasic arousal on decision-making. One account posits that threshold crossing of the decision variable triggers phasic noradrenaline release, facilitating the translation of the decision into a behavioural response[11]. In contrast to our observations, this framework focuses on functional effects of phasic arousal within the same trial, rather than subsequent ones, and it predicts improvements in sensitivity and/or RT[55], rather than changes in bias. Other accounts have proposed that phasic noradrenaline release facilitates a 'network reset'[56], enabling the transition of neural decision circuits to a new state[8]. Our group-level finding that high pupil-linked arousal reduces serial biases might be interpreted as the discarding of post-decisional activity traces due to network reset[57,58]. However, our analysis of individual choice patterns revealed that pupil-linked arousal boosted alternation also in those observers who already exhibited a tendency to alternate their choices, which is not easily reconciled with the network reset idea.

Previous theories of arousal and neuromodulation have coarsely distinguished between two timescales of arousal fluctuations: tonic fluctuations over the course of seconds to hours, and phasic responses on a sub-second timescale, time-locked to rapid cognitive acts[7,8,11]. Changes in tonic arousal occur spontaneously[13,59], and might also track changes in task utility or uncertainty[7,9–11]. Pupil-linked changes in tonic arousal strongly shape the operating mode of cortical circuits, including early sensory cortices, on slow timescales[13]. Phasic pupil-linked arousal responses, on the other hand, predict behaviour related to the same transient cognitive processes that drive them[29,42,60]. The uncertainty-linked pupil responses we identified here built up slowly after choice and predicted choice behaviour several seconds later. Thus, our current results suggest that pupil-linked arousal systems are driven by, and interact with, cognitive processes also at intermediate timescales; faster than tonic arousal, but more sustained than task-evoked phasic responses.

The dissociation between pupil- and RT-linked modulatory effects (Fig. 5f and Supplementary Fig. 11) on serial choice bias suggests that decision uncertainty signals were propagated along distinct central neural pathways, one linked to pupil responses and the other to RT, which then shaped serial choice biases in different ways. Even if the same uncertainty signals fed into these pathways, they might have become decoupled through independent internal noise. Specifically, it is tempting to speculate that the pupil-linked alternation boost reflected neuromodulator release from brainstem centres (such as noradrenaline from the LC[58]), whereas RT-linked bias reduction was driven by frontal cortical areas involved in explicit performance monitoring and top-down control (such as anterior cingulate cortex)[36,61,62]. Top-down effects of prefrontal cortex on decision-making[36,63] are commonly associated with explicit strategic effects that are adaptive within the experimental task. Indeed, the RT-linked modulation of serial bias was adaptive, in that it generally reduced observers' intrinsic serial bias. By contrast, pupil-linked arousal modulated serial choice patterns in a way that was maladaptive for part of the observers (the alternators). This finding might be related to the observation that maladaptive serial choice biases remain prevalent even in highly trained observers who know the statistics of the task[32,33]. Taken together, the dissociation between pupil- and RT-linked effects suggest that serial choice biases result from a complex interplay between low-level, pupil-linked arousal systems and higher-level systems for strategic control. Future studies should pinpoint the neural systems underlying these distinct effects, as well as their interactions[58].

In conclusion, our study identified decision uncertainty as a high-level driver of phasic arousal, and it uncovered a role of this pupil-linked arousal response in shaping the dynamics of serial choice biases—a pervasive but often ignored characteristic of human decision-making. These insights shed new light on the link between decision uncertainty, pupil-linked arousal state, and serial dependencies in decision-making. They set the stage for further investigations into the neural bases of arousal-dependent modulations of serial choice behaviour.

## Methods

**Operationalizing decision uncertainty.** In signal detection theory, a decision variable $dv_i$ is drawn on each trial from a normal distribution $N(\mu, \sigma)$ with $\mu$ corresponding to that trial's sensory evidence and $\sigma$ reflecting the internal noise. In Fig. 1, we used the range of single-trial motion energy values $[-6, 6]$ as our $\mu$. We estimated $\sigma$ from the data using a probit psychometric function fit on data combined across observers. The probit slope $\beta = 0.367$, where its inverse $\sigma = 2.723$ reflected the standard deviation of the dv distribution. The decision bound $c$ was set to 0, reflecting an observer without overall choice bias. The two pairs of distributions in Fig. 1 were generated using $\mu = -1$ and $\mu = 1$ for weak evidence, and $\mu = -4$ and $\mu = 4$ for strong evidence. To calculate the relationship between evidence strength and decision uncertainty (Fig. 1c), we simulated a normal distribution of dv for each level of evidence strength, with $\mu = [0,6]$ and $\sigma = 2.723$.

Since these uncertainty computations are symmetrical with respect to choice identity, we visualized only the pattern corresponding to $\mu > 0$ (stimulus B in Fig. 1a). All samples from such a distribution were split into correct and error parts based on their position with respect to the decision bound $c$. For each combination of evidence strength and choice, the average uncertainty level is

$$\text{Uncertainty} = 1 - \frac{1}{n} \times \sum_{i=1}^{n} f(|dv_i| - c) \tag{1}$$

where $f$ is the cumulative distribution function of the normal distribution

$$f(x) = \frac{1}{2}\left[1 + \text{erf}\left(\frac{x}{\sigma\sqrt{2}}\right)\right] \tag{2}$$

which transforms the distance between dv and $c$ into the probability of a correct response[21].

We simulated ten million trials based on the range of evidence in the data, and for each we computed a binary choice, the corresponding level of decision uncertainty, and the accuracy of the choice. Figure 1c–e visualizes the relationship between evidence strength, uncertainty and choice accuracy in these simulated data.

**Participants and sample size.** Twenty-seven healthy human observers (10 male, aged $23 \pm 5.2$ years) participated in the study. The ethics committee at the University of Amsterdam approved the study, and all observers gave their informed consent. We included all observers in each analyses presented in the paper. Each observer participated in one practice session and five main experimental sessions, each of approximately two hours and comprising 500 trials of the task. The number of observers was selected to allow for robust analyses of individual differences, as in previous pupillometry work from our laboratory[29], and the total number of trials per observer was chosen to allow for robust psychometric function fits and detection of subtle changes in the fit parameters.

**Task and procedure.** Observers performed a two-interval forced choice motion coherence discrimination task at constant luminance (Fig. 2a). Observers judged the difference in motion coherence between two successively presented random dot kinematograms (RDKs): a constant reference stimulus (70% motion coherence) and a test stimulus (varying motion coherence levels specified below). The intervals before, in between, and after (until the inter-trial interval) these two task-relevant stimuli had variable duration (numbers in Fig. 2a) and contained incoherent motion. A beep (50 ms, 440 Hz) indicated the onset of each (test and reference) stimulus. After offset of the test stimulus, observers had 3 s to report their judgment (button press with left or right index finger, counterbalanced across observers). After a variable interval (1.5–2.5 s), a feedback tone was played (150 ms, 880 or 200 Hz, feedback-tone mapping counterbalanced across observers). Dot motion was stopped 2–2.5 s after feedback, with stationary dots indicating the inter-trial interval, during which observers were allowed to blink their eyes. Observers self-initiated the next trial by button press (range of median inter-trial intervals across observers: 0.68–2.05 s).

The difference between motion coherence of test and reference was taken from three sets: easy (2.5, 5, 10, 20, 30), medium (1.25, 2.5, 5, 10, 30) and hard (0.625, 1.25, 2.5, 5, 20). All observers started with the easy set. We switched to the medium set when their psychophysical thresholds (70% accuracy defined by a cumulative Weibull fit) dropped below 15%, and to the hard set when thresholds dropped below 10%, in a given session.

Motion coherence differences were randomly shuffled within each block. We applied a counterbalancing scheme ensuring that within a block, each stimulus category (s2 > or < s1) was followed by itself or its opposite equally often[64]. The algorithm generated sequences of 53 trials, of which the first 50 were used per block.

**Random dot kinematograms.** Stimuli were generated using PsychToolbox-3 (ref. 65) and presented on a 22″ CRT monitor with a resolution of $1024 \times 768$ pixels and a refresh rate of 60 Hz. A red 'bulls-eye' fixation target[66] of 1.5° diameter was present in the centre of the screen. Dynamic random noise was presented in a central annulus (outer radius 12°, inner radius 2°) around fixation. The annulus was defined by a field of dots with a density of 1.7 dots/degrees², resulting in 768 dots on the screen in any given frame. Dots were 0.2° in diameter and had 100% contrast from the black screen background. All dots were divided into 'signal dots' and 'noise dots', whose proportions defined the varying motion coherence levels. Signal dots were randomly selected on each frame, and moved with $11.5° \text{ s}^{-1}$ in one of four diagonal directions (counterbalanced across observers). Signal dots that left the annulus wrapped around and reappeared on the other side. Signal dots had a limited 'lifetime' and were re-plotted in a random location after being on the screen for four consecutive frames. Noise dots were assigned a random location within the annulus on each frame, resulting in 'random position' noise with a 'different' rule[67]. Three independent motion sequences were interleaved[68], making the effective speed of signal dots in the display $3.8° \text{ s}^{-1}$.

**Motion energy filtering.** Due to the stochastic nature of the dynamic RDKs, the sensory evidence fluctuated within and across trials, around the nominal motion

coherence level. To quantify behaviour and pupil responses as a function of the actual, rather than the nominal, single-trial evidence, we used motion energy filtering to estimate those fluctuations[27].

Two spatial filters, resembling weighted sinusoids in opposite phase, were defined by

$$f_1(x, y) = \cos^4(a)\cos(4a)\exp\left(\frac{-y^2}{2\sigma_g^2}\right) \tag{3}$$

$$f_2(x, y) = \cos^4(a)\sin(4a)\exp\left(\frac{-y^2}{2\sigma_g^2}\right) \tag{4}$$

where $a = \tan^{-1}(x/\sigma_c)$. The parameters $\sigma_g = 0.05$ and $\sigma_c = 0.35$ defined the carrier sinusoid and the Gaussian envelope, respectively, in line with the response properties of MT neurons[69]. The coordinate system $(x, y)$ was rotated to match the stimulus' target direction or its 180° opposite. Two temporal filters were defined by

$$g_1(t) = (kt)^{n_s}\exp(-kt)\left[\frac{1}{n_s!} - \frac{(kt)^2}{(n_s+2)!}\right] \tag{5}$$

$$g_2(t) = (kt)^{n_f}\exp(-kt)\left[\frac{1}{n_f!} - \frac{(kt)^2}{(n_f+2)!}\right] \tag{6}$$

where $k = 60$ reflected the envelope of the temporal filters, and $n_s = 3$ and $n_f = 5$ controlled the width of the slow and fast filters, respectively[69]. A pair of spatio-temporal filters in quadrature pair was obtained by $f_1g_1 + f_2g_2$ and $f_2g_1 - f_1g_2$. We convolved each filter with the single-trial random dot movies. The resulting values were squared, and summed together across the two filters[27].

This filtering procedure was performed for each observer's individual target direction as well as its 180° opposite. To avoid cardinal biases in motion perception, we used the four diagonals as target directions counterbalanced across observers. Outputs of the two filtering operations were subtracted to yield a direction-selective signal over time[69]. To obtain a single measure of sensory evidence per trial, we averaged overall timepoints within each stimulus interval, and took the difference between motion energy in the first and second interval as our measure of single-trial sensory evidence. Evidence strength was defined by taking the absolute value of this sensory evidence, collapsing over the two stimulus identities (Fig. 2b).

**Pupillometry.** Observers sat in a dark room with their head in a chinrest at 50 cm from the screen. Horizontal and vertical gaze position, as well as the area of the pupil, were monitored in the left eye using an EyeLink 1000 desktop mount (SR Research, sampling rate: 1,000 Hz). The eye tracker was calibrated before each block of 50 trials.

Missing data and blinks, as detected by the EyeLink software, were padded by 150 ms and linearly interpolated. Additional blinks were found using peak detection on the velocity of the pupil signal and linearly interpolated. We estimated the effect of blinks and saccades on the pupil response through deconvolution, and removed these responses from the data using linear regression using a procedure detailed in ref. 70. The residual pupil time series were bandpass filtered using a 0.01–10 Hz second-order Butterworth filter, z-scored per run, and resampled to 100 Hz. We epoched trials, and baseline corrected each trial by subtracting the mean pupil diameter 500 ms before onset of the reference stimulus.

We included all trials from all five main sessions (that is, excluding the practice session) in the analyses reported in this paper. The time series of consecutive trial-wise stimuli, choices, RTs and pupil responses were necessary for the regression model of serial bias modulation. Observers were well-practiced in the task structure after the practice session. As a consequence, they made few blinks during the trial intervals (on average across observers, only 7.7% of trials contained more than 50% interpolated samples). The percentage of interpolated trials did not correlate with the estimated effect of pupil responses on serial choice bias ($r = -0.268$, $P = 0.175$, $\text{Bf}_{10} = 0.369$).

**Quantifying pupil time courses and single-trial responses.** To characterize the time-course of uncertainty encoding in the pupil response, we regressed across-trial evidence strength onto each sample of the baseline-corrected pupil signal, separately for correct and error trials (Fig. 3b). The design matrix for this regression also included an intercept and three nuisance covariates: (i) log-transformed RTs; (ii) sample-by-sample horizontal gaze coordinates; and (iii) sample-by-sample vertical gaze coordinates. We tested the significance of this regression time course using cluster-based permutation statistics[71].

We took the mean baseline-corrected pupil signal during 250 ms before feedback delivery as our single-trial measure of pupil response. Because of the temporal low-pass characteristics of the sluggish peripheral pupil apparatus[72], trial-to-trial variations in RT can cause trial-to-trial in pupil responses, even in the absence of amplitude variations in the underlying neural responses. To specifically isolate trial-to-trial variations in the amplitude (not duration) of the underlying neural responses, we removed components explained by RT via linear regression

$$\mathbf{y}' = \mathbf{y} - (\mathbf{y}^T\mathbf{r})\mathbf{r} \tag{7}$$

where **y** was the original vector of pupil responses, **r** was the vector of the corresponding single-trial RTs (log-transformed and normalized to a unit vector), and $T$ denoted matrix transpose. The residual **y**′ thus reflected pupil responses, after removing variance explained by trial-by-trial RTs. This residual pupil response was used for all analyses reported in the main text.

**Quantifying post-error slowing.** We quantified post-error slowing, for tertiles of previous trial pupil responses, as described in ref. 30. Briefly, we selected those error trials that were both preceded and followed by a correct trial, and subtracted the pre-error RT from the associated post-error RT. This procedure ensured that estimates of post-error slowing could not be driven by error-unrelated, intrinsic fluctuations in RT over the course of a session. Before this subtraction, we removed trial-by-trial evidence strength from RTs using linear regression, to account for the large variations in RT with stronger sensory evidence (Fig. 2d).

**Quantifying the psychometric function.** We modelled the psychometric function (Supplementary Fig. 4a) as follows. The probability of a particular response $r_t = 1$ on trial t was described as

$$P(r_t = 1 | \widetilde{s_t}) = \gamma + (1 - \gamma - \lambda)\, g(\delta + \alpha \widetilde{s_t}) \qquad (8)$$

where $\lambda$ and $\gamma$ were the probabilities of stimulus-independent errors ('lapses'), $\widetilde{s_t}$ was the signed stimulus intensity (here: signed sensory evidence as in Fig. 2b), $g(x) = 1/(1 + e^{-x})$ was the logistic function, $\alpha$ was perceptual sensitivity, and $\delta$ was a bias term. The free parameters $\gamma$, $\lambda$, $\alpha$ and $\delta$ were estimated by minimizing the negative log-likelihood of the data (using Matlab's fminsearchbnd). We constrained $\gamma$ and $\lambda$ to be identical, so as to estimate a single, choice-independent lapse rate.

For the quantification of serial choice bias (Supplementary Fig. 5), we binned the data by previous choices and by previous pupil responses or RT. For each of those subsets of trials, we fit the psychometric function (equation (8)) to choices on the subsequent trials. The resulting bias term $\delta$ was transformed from log-odds into probability by $P = e^{\delta}/(1 + e^{\delta})$. This quantified $P(r_t = 1)$ for ambiguous evidence (that is, strength of zero). Collapsing these values across the two-choice options (shown separately in Supplementary Fig. 5) yielded the pooled measure of choice repetition probability in Fig. 4a,f.

**Quantifying perceptual sensitivity using cumulative Weibull function fits.** In Fig. 3d and Supplementary Figs 1c and 3b, we fit a cumulative Weibull function to accuracy as a function of evidence strength. The probability of a correct response $c_t = 1$ on trial t was defined as

$$P(c_t = 1 | s_t) = 1 - (0.5 - \lambda)\, f\!\left(\left(\frac{s_t}{\theta}\right)^{\beta}\right) \qquad (9)$$

where $s_t$ was the absolute evidence strength, $f(x) = (1 - e^{-x})$ was the cumulative Weibull function, $\lambda$ was the lapse rate, $\theta$ was the threshold indicating at which level of evidence strength an accuracy of ~80% is achieved, and $\beta$ was the slope of the cumulative Weibull function. The free parameters $\theta$, $\beta$ and $\lambda$ were estimated by minimizing the negative log-likelihood of the data (using Matlab's fminsearchbnd). Perceptual sensitivity was then defined as $1/\theta$.

**Modelling the modulation of serial choice bias.** We modelled the pupil- and RT-linked modulation of serial choice bias by extending an established regression approach[33]. The basic regression model extended the psychometric function model from equation (8) by means of a history-dependent bias term $\delta_{\mathrm{hist}}(\mathbf{h}_t)$, which was a linear combination of previous stimuli and choices

$$P(r_t = 1 | \widetilde{s_t}, \mathbf{h}_t) = \gamma + (1 - \gamma - \lambda)\, g(\delta(\mathbf{h}_t) + \alpha \widetilde{s_t}) \qquad (10)$$

With

$$\delta(\mathbf{h}_t) = \delta' + \delta_{\mathrm{hist}}(\mathbf{h}_t) = \delta' + \sum_{k=1}^{K} \omega_k h_{kt} \qquad (11)$$

where the bias term $\delta(\mathbf{h}_t)$ was the sum of the overall bias $\delta'$ (see equation (8)) and the history bias $\delta_{\mathrm{hist}}(\mathbf{h}_t) = \sum_{k=1}^{K} \omega_k h_{kt}$, where $\omega_k$ were the weights assigned to each previous stimulus or choice. We here modelled

$$\mathbf{h}_t = (r_{t-1}, r_{t-2}, r_{t-3}, r_{t-4}, r_{t-5}, r_{t-6}, r_{t-7}, z_{t-1}, z_{t-2}, z_{t-3}, z_{t-4}, z_{t-5}, z_{t-6}, z_{t-7}) \qquad (12)$$

as a concatenation of the last seven responses and stimuli (see ref. 33 for details). This procedure allowed us to quantify the effect of past trials on current choice processes (Fig. 5c). We convolved every set of seven past trials with three exponentially decaying basis functions[33]. Positive history weights $\omega_k$ indicated a tendency to repeat the previous choice, or to make a choice that matched the previous stimulus. Negative weights described a tendency to alternate the corresponding history feature.

To model the effect of pupil-linked uncertainty on history biases, we extended this model by adding a multiplicative interaction term $\sum_{k=1}^{K} \omega'_k h_{kt} p_{kt}$, which described the interaction of pupil responses with the choice and stimulus identities

at the last seven lags:

$$P(r_t = 1 | \widetilde{s_t}, \mathbf{h}_t, \mathbf{p}_t) = \gamma + (1 - \gamma - \lambda)\, g(\delta(\mathbf{h}_t, \mathbf{p}_t) + \alpha \widetilde{s_t}) \qquad (13)$$

$$\delta(\mathbf{h}_t, \mathbf{p}_t) = \delta' + \delta_{\mathrm{hist}}(\mathbf{h}_t, \mathbf{p}_t) = \delta' + \sum_{k=1}^{K} \omega_k h_{kt} + \omega'_k h_{kt} p_{kt} + \omega''_k p_{kt} \qquad (14)$$

where $\omega'_k$ were the history × pupil interaction weights, $\omega''_k$ were the pupil weights and $p_{kt} = (p_{t-1}, p_{t-2}, p_{t-3}, p_{t-4}, p_{t-5}, p_{t-6}, p_{t-7})$ was a concatenation of the last seven pupil responses. The term $\omega''_k p_{kt}$ acted as a nuisance covariate. To simultaneously model the effects of pupil responses and log-transformed RT, our model also included RT and history × RT terms, generated using the same procedure.

All parameters were fit using an expectation maximization algorithm. To assess whether individual observers were significantly influenced by their experimental history, we ran 1,000 iterations of permuting all trials, fitting the full model, and subsequently comparing the likelihood of the intact model to this null distribution (where permutation nullifies true history effects)[33]. Confidence intervals for individual regression weights were obtained from a bootstrapping procedure.

**Serial bias and outcome-dependent choice strategies.** The history weights for past stimuli and responses allowed us to characterize different decision strategies[33] (Fig. 5d). Positive weights associated with the previous choice, or the previous stimulus category, indicate a tendency to repeat this previous choice, or to make a choice corresponding to the previous stimulus, respectively. Negative weights correspond to a tendency to alternate previous choice or stimulus. In the left and right triangle of this strategy space, the magnitude of the response weight is larger than the magnitude of the stimulus weight. Consequently, strategies are dominated by the previous choice and can be simply defined as choice alternation (left) or choice repetition (right).

In the upper and in the lower triangle, the magnitude of the stimulus weight is larger than the magnitude of the response weight, so strategies are dominated by the identity of the previous stimulus (which is only known to the observer as a function of their previous response and feedback). In the upper and lower triangle, strategies are thus defined by the sign of the stimulus weight. In the upper triangle stimulus weights are positive, indicating a tendency to repeat the previous stimulus. On a correct trial, previous choice and stimulus are equal and therefore, repeating the previous stimulus is equal to repeating the previous choice (a win-stay strategy). On errors, the previous choice is opposite to the previous stimulus and repeating the previous stimulus is equal to alternating the previous choice (lose-switch strategy). Conversely, in the lower triangle stimulus weights are negative, reflecting a tendency to alternate the previous stimulus. This implies a tendency to alternate the previous choice if the previous choice was correct (win-switch strategy) and a tendency to repeat the previous choice in case of a previous error (lose-stay strategy).

The weights for previous choices and stimuli can easily be combined to obtain weights reflecting a tendency to repeat previous correct or incorrect choices (Supplementary Fig. 6). Specifically, correct weights are defined by choice + stimulus, and error weights by choice—stimulus[33]. The same holds for modulation weights. This transformation is identical to fitting a model with regressors for previous successes and failures[39,73].

**Statistical tests.** We used non-parametric permutation testing to test for the group-level significance of individually fitted parameter values (Figs 3 and 5e,g). We randomly switched labels of individual observations either between two paired sets of values, between one set of values and zero, or between two unpaired groups. After repeating this procedure 10,000 times, and computing the difference between the two group means on each permutation, the P value was the fraction of permutations that exceeded the observed difference between the means. All P values reported were computed using two-sided tests.

In Fig. 4, we split the data into tertiles of pupil response or RT, and computed next trial serial choice bias, overall choice bias, signed choice bias, perceptual sensitivity, lapse rate, RT and post-error slowing in each bin. We used a repeated-measures ANOVA to test for the main effect of bin on each dependent variable. We further used Bayes Factors (Bf), obtained from a Bayesian one-factor ANOVA[74], to support conclusions about null effects observed. $\mathrm{Bf}_{10}$ quantifies the evidence in favour of the null or the alternative hypothesis, where $\mathrm{Bf}_{10} < 1/3$ or $> 3$ is taken to indicate substantial evidence for $H_0$ or $H_1$, respectively. $\mathrm{Bf}_{10} = 1$ indicates inconclusive evidence. We similarly computed $\mathrm{Bf}_{10}$ for correlations, based on the Pearson correlation coefficient[75].

The P-value for the difference between the two correlation coefficients (choice weight by pupil modulation weight vs choice weight by RT modulation weight), shown in Fig. 5f, was obtained through permutation testing. To generate a null distribution of no difference, we randomly switched (or not, dependent on a simulated coin flip) each observer's RT and pupil modulation weights, after which we computed the between-subject correlation between choice weights and pupil modulation weights as well as between choice and RT modulation weights. Repeating this procedure 10,000 times generated a distribution of the difference in correlation, under the null hypothesis of no difference.

**Data availability.** All raw and processed data, as well as the code to reproduce all analyses and figures, are available at http://dx.doi.org/10.6084/m9.figshare.4300043.

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

## Acknowledgements

We thank O'Jay Medina for assistance with data collection, all members of the Donnerlab for valuable discussions, and Konstantinos Tsetsos, Jan Willem de Gee, Niklas Wilming, Camile Correa, Florent Meyniel and Sander Nieuwenhuis for helpful comments on the manuscript. We acknowledge computing resources provided by NWO Physical Sciences.
This research was supported by the German Academic Exchange Service (DAAD) and G.-A. Lienert Foundation (to A.E.U.) and the German Research Foundation (DFG), SFB 936/A7, SFB 936/Z1, DO 1240/2-1 and DO 1240/3-1, and European Union Seventh Framework Programme (FP7/2007-2013) under grant agreement no. 604102 (Human Brain Project) (to T.H.D.).

## Author contributions

Conceptualization, A.E.U. and T.H.D.; Investigation, A.E.U.; Formal Analysis, A.E.U. and A.B.; Software, data curation and visualization, A.E.U.; Writing, A.E.U. and T.H.D.; Supervision, T.H.D.

## Additional information

**Competing financial interests:** The authors declare no competing financial interests.

