## [Peer Review File · Nature Communications]

Reviewers' Comments:

Reviewer #1 (Remarks to the Author):

The manuscript by Urai and colleagues quantitatively examines how post-decisional pupil dilation can serve as a measure of confidence that predicts upcoming choice patterns. In psychology confidence has been often associated with higher order cognition and clearly has the potential to impact choices. Yet, it has been difficult to study confidence and the field has mainly relied on introspective reports that raised a number of issues. Here the authors use a formal framework to establish that pupil diameter after a choice can serve as a proxy for confidence, defined as the probability of making a correct choice. Then they go on to show that pupil diameter predicts systematic choice biases. The experiments were designed carefully and I found the analysis and data presentation elegant. Overall, I enjoyed this paper and I think it has the potential to have a significant impact on the field.

I have two main lines of comments to the authors that I think should be addressed before this paper is published. First, the authors often use imprecise language to describe what they have done and this unnecessarily inflates their claims. Second, they need to perform some further analysis to support the claims that pupil diameter is a good proxy for confidence.

(1) The authors study choice patterns and pupil diameter yet make a number of unwarranted claims about arousal and neuromodulators and under potential neural systems underlying their results. I find these distracting and unnecessary.

For in instance the abstract concludes with "We conclude that decision uncertainty drives rapid changes in central arousal state, shaping the serial correlation structure of choice behaviour." I am not sure how they can make claims about "arousal state" in general.

Other examples, like "We hypothesised that decision-makers' uncertainty about their preceding choice shapes serial choice biases by recruiting the arousal systems of the brainstem. These systems are in an ideal position to transform decision uncertainty into adjustments of behaviour: They receive descending projections from frontal brain regions encoding decision uncertainty and have widespread modulatory effects on their cortical target networks."

This is a very nice hypothesis but the authors' work has no bearing on it. Pupil dilation will correlate with numerous brain systems and the origins of the signal or its causality are not studied.

"We applied motion energy filtering³⁰ to the stochastic random dot motion stimuli to compute an estimate of each trial's sensory evidence, which approximates the population response of motion-selective neurons in visual cortical area MT³¹"

I think the authors compute the average motion energy which is the not a good measure of what MT neurons encode. Please leave the speculation out of the Results section.

"To this end, we analysed reaction time (RT), an established proxy for decision uncertainty^{29,32,33} ."

RT is a proxy for just about every type of behavioral measure. The observations are fascinating but I wouldn't call RT an established proxy for uncertainty.

There are several other such claims, please remove them from the Introduction and Results section. The Discussion of course the authors are free to speculate about their results might be relevant for neural systems.

(2) Throughout the paper the authors rely on a single signature of confidence, as shown in Figure 1c. Although I agree that it is the most useful, somewhat counterintuitive signature, I think the authors should try to establish all three known signatures that follow from the signal detection model (Kepecs et al, 2008). Perhaps the most relevant presentation of these are in a recent paper by Sanders et al, 2016. In particular, it would be important to establish how well pupil diameter predicts choice accuracy (Sanders paper Fig 1B) and whether this prediction is beyond what can be made by using decision difficulty on its own (Sanders paper Fig 1D).

Importantly, the authors need show in Figure 2, for RT, that the prediction of accuracy is not below 50% thus showing that it is not "error detection" but confidence. The same applies to the pupillometry data in other figures. From these two signatures the third might follow but ideally the authors should directly show that the psychometric slope is different for high vs. low pupil diameter.

(3) If I understand the history analysis correctly the authors separately examine stimulus identity and previous choice but do not have a regression on previous outcome. My question is how much does a term in Previous choice X previous outcome affects the regression? Figure 4d does seem to suggest that there is a correlation between choice weight and stimulus weight indicating perhaps an interaction between stimulus and choice. Similarly in Figure 6, the analysis could be done using the previous outcome.

Reviewer #2 (Remarks to the Author):

The manuscript reports a tremendously rigorous and detailed analysis of sequential dependencies in a visual motion discrimination task and their relation to arousal as measured through pupil dilation. The authors demonstrate that post-decision pupil dilation scales with levels of uncertainty inferred from an SDT model of decision uncertainty and from experimentally observed RTs. The size of the pupil response predicted a shift to choice alternation in the

perceptual task.

The main novelty of the study is linking uncertainty-related arousal to sequential dependencies. The other main finding, that arousal increases with uncertainty and errors, is now established using measures including pupil dilation and heart rate changes: Lempert et al. (2015, PLoS One), Wessel et al. (2011, J Cogn Neurosci), Hajcak et al. (2003, Psychophysiology). However, a weakness of the main novel result is that the causality is not established. It is not intuitive that increased arousal should result in choice alternation, and the manuscript does not develop a strong case for this prediction (the relevant analysis seems exploratory rather than hypothesis driven). Equally, there are many potential routes by which frontally-coded uncertainty could affect choice, some of which seem better established or more plausible than the one proposed. Why not a direct effect within frontal cortex (Botvinick et al., 2001, Psych Rev), or between frontal cortex and basal ganglia? One recent suggestion is that confidence can guide perceptual learning by augmenting well-known reinforcement learning processes (Guggenmos et al., 2016, eLife), which could produce the effects reported here.

The manuscript is otherwise clear and persuasively written, with good treatment of uncertainties and confounds (e.g. effects of difficulty). That said, I found the treatment of the RT/uncertainty/dilation relationship confusing in places. The RT analysis in Figure 1 is first interpreted to establish the effectiveness of the manipulation of decision uncertainty, but in the next section the weak relationship between RT and pupil responses is also interpreted positively. Could the authors clarify their thinking about the relationship between RT, uncertainty and pupil responses? Tangentially related, to what extent is the difference between pupil response's and RT's relationship with choice weight an artifact of removing log-transformed RT from the trial-to-trial fluctuations of pre-feedback pupil responses via linear regression?

As mentioned above, the analysis is commendably thorough. One aspect that I did not see discussed is the extent to which strength of serial choice biases varied with the difficulty of the current trial. Presumably biases were stronger on more difficult trials, where choices would be less dominated by sensory evidence?

Partly related to this point, p.14 discusses effects of small serial correlations between stimuli. Did the authors also rule out possible effects of serial correlations between difficulty levels? From the description it seems possible that low coherence trials would tend to be followed by higher coherence trials and vice versa. Could this dependency, if present, contribute to the apparent effects of uncertainty on serial choice biases, as measured either through pupil responses or RT?

Reviewer #3 (Remarks to the Author):

Urai and colleagues expand on a mounting body of work linking arousal processes to the underlying computations that govern decision-making. The authors measure pupil diameter in a fixed-duration two-interval motion comparison task where trial-by-trial evidence can be estimated through motion energy filtering. The authors develop qualitative predictions for decision uncertainty across task conditions and show that both reaction time and pre-feedback pupil diameter follow the predicted patterns. Each of these variables also predicts the extent to which trial choice will bias behavior on the subsequent trial, with larger pupil diameters and longer reaction times corresponding to smaller average choice biases across subjects. Interestingly, this group effect seems to come about for different reasons in the two variables: while longer reaction time related to lower overall levels of sequential dependencies irrespective of the sign (eg. perseveration versus alternation), pupil diameter seemed to increase alternation behavior, even in subjects for whom this sort of choice bias was dominant.

I have been asked to address the novelty of the findings of this paper, particularly in light of other recent papers on similar topics (eg. 1,2). While the basic idea that pupil diameter reflects an uncertainty signal has been shown before in slightly different ways (eg. 2,3), in my opinion the effects of this signal on sequential dynamics are both novel and important (particularly given 4). To the best of my knowledge the only paper that touches on this concept is 3 (since prior choice/belief dependencies are reduced as learning rate goes to one), but the current paper offers several key advances: 1) the authors show that the effect is not just a reduction of perseveration and in some cases enhances existing alternation biases, 2) the authors quantify the specific linkage between the uncertainty signal and sequential dependencies, and 3) the authors show the effect in a commonly used perceptual task for which sequential dependencies are sub-optimal yet pervasive. Overall I see the results as timely, interesting and generally well communicated. Nonetheless, there are several concerns that I have with the manuscript in its current form:

Major concerns:

The authors provide a very concise narrative to describe the key findings, but in places I think it would have been useful to have some more basic analysis. In particular, choice or accuracy data would have been useful in evaluating the evidence strength metrics provided in figure 2. In addition, since the data were collected across multiple sessions, it would be good to know how performance (accuracy & RT) change within and across sessions and to ensure that the key findings are not simply due to pupil and behavior both changing as a function of time/training. The authors should also include some analysis of how choice history curves (figure 4) change over training. Given that history was not predictive of stimulus, I would expect that these effects shrink over time. Any such effects should be accounted for in the pupil and RT analyses of

sequential bias (figure 5).

It would be useful if the authors could provide example RT distributions and comment on the level of overall variability in RTs. In particular, it was not clear to me whether the reduced sequential dependencies for long RT trials could simply be due to the additional passage of time.

The authors control for RT in assessing the effects of pupil diameter in sequential bias, however, as far as I can tell, they did not control for previous trial accuracy, which explains a large proportion of the variance in the pupil data. It would be good to know whether trial accuracy itself affects sequential bias and to ensure that the effects of pupil diameter are not completely mediated by accuracy differences.

One lingering question in my mind is whether the relationship between alternation-type sequential dependencies and pupil diameter occurs over a short timescale (eg. trial-to-trial) or whether they go through slow and synchronized fluctuations. To this effect it would be useful if the authors could conduct their analyses (eg. Fig 6 B) on time shifted data to, for example, determine whether the pre-feedback pupil response on trial t has any relationship to the sequential biases observed for trials t or $t+2$.

Page 10, paragraph 2, the authors say that: "the bias component of choices became less history-dependent, more neutral and thereby better adapted to the statistics of the task", but later on, they look more closely and show that in reality, this was only true for some subjects. In particular, the subjects that showed alternation biases were more biased on trials following a large pupil response. It would be interesting to know whether this basic idea bears out by looking at how accuracy relates to pupil diameter for perseverative versus alternating subjects.

The authors discount the idea that conditional (error/correct) changes in RT over time could be due to task strategy "since the stimulus duration was fixed, observers could not accumulate more evidence by slowing down decisions on uncertain trials." However, I am not totally convinced by this. Couldn't subjects make a decision during motion viewing on some trials leading to action preparation during the 500 ms viewing interval and faster RTs? Once again, knowing what the RT distributions looked like might be informative in this regard.

The authors go on to discard the notion that mental effort could explain the linkage between pre-feedback pupil diameter and evidence (conditioned on accuracy). However, along the lines of my previous comment, you could imagine operationalizing mental effort as something like: "time accumulating information". With such a definition, and with an accumulation process incorporating bound separation variability and truncated by the stimulus offset, I could imagine that you might get a similar predicted pattern. This is particularly relevant giving that DDM modeling of value-based decision task behavior has suggested that arousal relates to an increased decision threshold, which would lead to a longer accumulation time. Not at all to say that this

detracts from the authors primary claim regarding the downstream effects of this signal on subsequent behavior, but I think that if the authors would really like to rule out mental effort they should consider more robust versions of it.

The authors claim that the effect of pupil response on subsequent behavior was specific for pre-feedback pupil responses based on a failure to find an effect in the post-feedback responses after regressing out shared variance, however it seems that supporting this claim would require the converse to be true. Otherwise the pupil measures may just be too highly correlated to see any effects in the unique variance.

It would be useful if the authors could provide more discussion of the linkage between the effects of arousal on sequential biases and relevant ideas regarding the role of arousal in learning and decision-making. In my mind, the authors finding is, in broad strokes, very much in line with two ideas... one from decision making suggesting that increased tonic NE can promote exploration by decreasing the weight of learned action values β and one based on learning, that NE can allow information learned in a now irrelevant regime to be discarded or omitted from use in forming future decisions β , γ . That said, the specifics of the findings may not fit exactly, as neither of those theories would account for the promotion of alternation. In any case, for the results of this study to be interpreted broadly, it would be useful for them to be cast in the context of existing theory.

Minor concerns:

Does the evidence conditioned on choice relationship to pupil diameter hold until the beginning of the subsequent trial? If so, is pupil diameter at the beginning of a trial predictive of the sequential bias that will be displayed on that trial?

Page 3, paragraph 2: "this quantity is encoded..."

This sentence could be read to mean that uncertainty is represented in one set of regions in monkeys and a different set of regions in rodents.

1. de Gee, J. W., Knapen, T. & Donner, T. H. Decision-related pupil dilation reflects upcoming choice and individual bias. *Proceedings of the National Academy of Sciences* 111, E618-E625 (2014).
2. Lempert, K. M., Chen, Y. L. & Fleming, S. M. Relating Pupil Dilation and Metacognitive Confidence during Auditory Decision-Making. *PLoS ONE* 10, e0126588 (2015).
3. Nassar, M. R. et al. Rational regulation of learning dynamics by pupil-linked arousal systems.

Nature Neuroscience 15, 1040-1046 (2012).

4. Tervo, D. G. R. et al. Behavioral Variability through Stochastic Choice and Its Gating by Anterior Cingulate Cortex. *Cell* 159, 21-32 (2014).

5. Cavanagh, J. F., Wiecki, T. V., Kochar, A. & Frank, M. J. Eye Tracking and Pupillometry Are Indicators of Dissociable Latent Decision Processes. *Journal of Experimental Psychology: General* (2014). doi:10.1037/a0035813

6. Aston-Jones, G. & Cohen, J. D. An integrative theory of locus coeruleus-norepinephrine function: adaptive gain and optimal performance. *Annu. Rev. Neurosci.* 28, 403-450 (2005).

7. Yu, A. J. & Dayan, P. Uncertainty, neuromodulation, and attention. *Neuron* 46, 681-692 (2005).

Reviewer #4 (Remarks to the Author):

Urai and colleagues present a timely and well-conducted demonstration that (perceptual) decision uncertainty modulates serial choice biases, possibly through changes in arousal state, here assessed through recordings of pupil dilatation. Using a model of decision uncertainty adapted from signal detection theory, the authors obtain predictions about the scaling of uncertainty with varying strength of sensory evidence, separately assessed for correct and incorrect trials. The authors recorded pupil diameter as a proxy for central arousal state, which follows some of their own elegant work and recent advances in linking pupil diameter to cognitive processes. The authors report that pupil diameter scales with decision uncertainty, after the choice is made but prior to presentation of feedback. The predominant behavioural bias across participants was for choice repetition, however, pupil responses explained a switch in choice alteration. The authors conclude that the encoding of uncertainty is dynamically updating behavioural responses, and that this can be read-out using dynamic tracking of pupil responses. The paper is well-written, and the methods are thorough and the work follows their own previous work that has contributed to establishing pupil responses as an important physiological readout that provides a window into the neural dynamics steering cognition. The paper itself contains a considerable amount of work, and the results appear sound, with several control analyses presented.

I have several questions which broadly pertain to novelty and clarification, in no specific order of relevance

- The paper is a nice extension of previous work, and aligns with reports that link changes in pupil diameter with uncertainty computations, and these in turn with behavioural changes. At the same time, it would be helpful to see the present work more directly discussed with other recent efforts to link uncertainty with pupil dynamics - how does the present work extend these other reports? (e.g. Peuschoff et al 2011; Nassar et al 2012; de Berker et al 2016; de Gee et al 2014)
- A related point relates to the specific type of decision uncertainty the authors here investigate

and quantify. How does this formally relate to other forms of uncertainty, as addressed in some of the papers mentioned above? This point seems relevant not just for relating it to other work, but also the issue of decision uncertainty, response conflict, and reward expectancy in the present study highly overlapping. Put simply, if these three quantities cannot be divorced, then is the construct validity of the paper limited? That is, if one agrees that these three quantities are related, the perhaps it would be useful to recognise earlier on in the paper that decision uncertainty is one of the possible variables that pupil diameter can reflect.

- What is the optimal behaviour in the present task? If I understand correctly, each stimulus category was followed by itself or its opposite alternative with equal probability. Why should we then expect a tendency for choice alteration under high decision uncertainty, and would this be in any way adaptive here? I guess this could be the case simply because there was a tendency for predominant choice bias across individuals, i.e. to repeat choices. If the case, perhaps the authors can explain this in more detail. Related, there is a long-standing literature on choice biases, recency effects and post-error slowing in the psychological literature. It would be good to be reassured how the present work relates to this earlier work. Also related, did the authors explicitly account for post-error slowing, and how is this related to pupil dilatation? Unless addressed in earlier work that I am not aware of, this could be a useful contribution to these lines of work.

Minor:

- I may have missed this, but what was the % of correct/incorrect trials? Did performance relate to their individual tendency to be a choice alternator, or a repeater?

- If I understand the task instruction correctly, then I think the term "reaction time" should be replaced by "response time", as there was no premium on responding quickly. In that case, some of the discussion, especially when relating to drift models and decision boundaries, may require revision. Similarly, it is then not surprising that no SAT was observed.

- On page 10, the authors state that post-feedback pupil responses predicted modulation of serial choice biases, but no longer did so when removing via linear regression variance explained by pre-feedback pupil responses from the post-feedback pupil signal. It may be worth pre-empting this analysis on page 6.

- From the task description I wasn't sure whether there was any change in stochasticity to learn and track from? If not, why would changes in learning rates be of use here?

- If I understand correctly, 8 participants did not show a tendency for choice alteration, and these were omitted from some analyses. But some clarification is warranted how and when these data were used - for example, if choice biases were not advantageous in the present task, then it seems more difficult to justify to exclude these participants, as would be, for example, if this was a learning experiment and these participants had simply not learned anything.

- Am I correct that the 7-back analysis is motivated by the previous paper by Frund? How does that analysis relate to the use of a "decay" function, in which a single parameter could determine how many past trials influence the current choice process?

- Would the authors expect any pupil responses to auditory beeps, as suggested in earlier work?

I wonder if this should be accounted for, and would take the author's guidance on this point. Related, would one expect a pupil response to the button press, and if so, is carry-over a possible concern?

- Could the authors report in more detail what the results of their artefact removal procedure was? For example, previous work has reported the occurrence of subjects with blinks etc requiring interpolation of >50%, which may warrant exclusion of those subjects (e.g. Browning et al 2015)

- Intro: "this quantity is encoded..." This sentence reads as if those regions are established as encoding decision uncertainty, with no other candidates. Is that so?

- On page 6, the last paragraph seems to engage in a strawman argument. Given quite a large number of pupil studies over the past years, would anyone still argue that pupil dilatation relates to "effort"? Besides, what does effort then actually mean, it seems a poorly operationalized term, perhaps a heuristic at best.

- Page 16: Do I understand correctly that the pupil signal was in essence quantified as the peak value at 250ms before feedback delivery? Can the authors explain how this account for any overlapping between different quantities that may influence pupil, e.g. beeps, uncertainty, stimulus offset etc? I may have missed this from the methods, in which case I would kindly ask the authors to point me to the respective sections.

NCOMMS-16-06414: Response to reviewers

Dear editors and reviewers,

We thank the reviewers for their constructive and helpful evaluation of our manuscript. We have addressed each of the reviewers' concerns, by performing a large number of additional analyses and by substantially revising all sections of the manuscript.

We believe that these major revisions have considerably strengthened our paper and the conclusions we can draw from the results. In what follows, we first provide a brief summary of our major revisions, followed by a detailed point-by-point reply to each of the reviewers' comments (printed in *blue italics*).

We hope that the reviewers will now find our manuscript suitable for publication in *Nature Communications*.

Sincerely,

Anne Urai, Anke Braun, and Tobias Donner

Summary of major revisions

First, we have substantially revised the complete text in response to comments from all reviewers. In response to reviewers #1 and #2 we have more clearly spelled out the specific rationale and hypotheses underlying our study; and we also have clarified and motivated our focus on "pupil-linked arousal" without making too many assumptions about the specific underlying neural substrates in the Introduction. Instead, we have added Discussion paragraphs (1) summarising a recent body of rigorous work establishing a close link between pupil diameter and cortical arousal state (which motivated our focus on pupil-linked arousal) and (2) speculating about possible neural pathways mediating our findings. In response to reviewers #3 and #4, we included new Discussion paragraphs on the relationship between our current findings and (1) existing theories of the functional role of arousal and (2) previous studies relating pupil dynamics to uncertainty.

Second, we have performed a number of new analyses (shown in several new panels of the main Figures and in 9 new Supplementary Figures). We extended Figures 1 and 3 to show all the signatures of decision uncertainty from Sanders et al. (2016), and the former Figure 5 (now: Figure 4) to show all components of decision-making (RT, sensitivity, lapse rate, absolute bias, and serial choice bias) as a function of previous trial pupil dilation. In line with the corresponding change in the presentation of the results in the main text – to clarify the development of the case for the specific pupil-dependent effects – we merged the former Figures 4 and 6 into the current Figure 5.

In addition to these changes in response to the reviewers' comments, we have made the following changes, to improve clarity and style. (1) We have changed the title of the manuscript from "Decision uncertainty drives pupil-linked arousal systems and alters subsequent decision-making" to "Pupil-linked arousal is driven by decision uncertainty and alters serial choice bias". (2) We now show the individual data points overlaid on the bar graphs representing the group mean. (3) We changed the colour schemes to red/blue for error/correct trials and added markers distinguished by shape, to improve legibility for colour-blind readers. (4) We have moved the parts of the Methods section to the Supplement to comply with the maximum word count and reference count.

Point-by-point reply to reviewer #1

The manuscript by Urai and colleagues quantitatively examines how post-decisional pupil dilation can serve as a measure of confidence that predicts upcoming choice patterns. In psychology confidence has been often associated with higher order cognition and clearly has the potential to impact choices. Yet, it has been difficult to study confidence and the field has mainly relied on introspective reports that raised a number of issues. Here the authors use a formal framework to establish that pupil diameter after a choice can serve as a proxy for confidence, defined as the probability of making a correct choice. Then they go on to show that pupil diameter predicts systematic choice biases. The experiments were designed carefully and I found the analysis and data presentation elegant. Overall, I enjoyed this paper and I think it has the potential to have a significant impact on the field.

Thank you for these kind words.

I have two main lines of comments to the authors that I think should be addressed before this paper is published. First, the authors often use imprecise language to describe what they have done and this unnecessarily inflates their claims. Second, they need to perform some further analysis to support the claims that pupil diameter is a good proxy for confidence.

The authors study choice patterns and pupil diameter yet make a number of unwarranted claims about arousal and neuromodulators and under potential neural systems underlying their results. I find these distracting and unnecessary.

Thank you for raising this important issue. In fact, we have debated this issue at length prior to the submission of our paper. On the one hand, we completely share the your desire to stay close to the data in the Introduction and Results sections; on the other hand, we feel it is crucial to convey, early on, to the reader why we chose to measure pupil diameter rather than any other physiological measure. Having re-assessed the manuscript, we agree that our initial version went too far in drawing direct links between our measurements and specific neural systems. We have now moved these speculative parts from the Introduction to the Discussion, where we elaborate on possible interpretations at a neural systems level (Discussion, 4th paragraph). We have also removed all references to specific neural pathways, brain regions, and neuromodulators, from the Results section.

In the Introduction, we now also define the specific focus of our study on pupil-linked components of cortical arousal, abbreviated by the term “pupil-linked arousal”. Rather than “arousal” as a general concept, this descriptive term has a clear operational definition. We now use this term throughout the Abstract, Introduction, and Results.

Our focus on pupil-linked arousal is based on recent work establishing a close link between pupil diameter and global cortical network state, ranging from single-neuron membrane potentials to LFPs and brain-wide patterns of fMRI responses (Eldar et al., 2013; McGinley et al., 2015; Pisauro et al., 2016; Reimer et al., 2014; Vinck et al., 2015; Yellin et al., 2015). For example, McGinley et al. (2015) report coherence between pupil diameter and hippocampal ripple rate of about 0.8 (their Figure 1E), and pupil diameter and neocortical membrane potential fluctuations of about 0.7 (their Figure 4). Vinck et al. (2015) report correlations of pupil diameter with LFP activity in delta (1-4 Hz) and gamma (55-65 Hz) frequency bands of close to 1 (their Figures 2C and 6B).

While these well established forebrain correlates of pupil diameter are important for motivating our study, our conclusions do not rely on any specific assumptions about the underlying neural

pathways (including ascending brainstem systems) causing the changes in global cortical state. We hope that you are satisfied with these revisions in general; we address your specific examples in the following.

For instance the abstract concludes with "We conclude that decision uncertainty drives rapid changes in central arousal state, shaping the serial correlation structure of choice behaviour." I am not sure how they can make claims about "arousal state" in general.

We agree that arousal is too broad a concept, and that we need to be more specific in our conclusions. As explained above, we have now rephrased all references to "arousal state" in terms of "pupil-linked arousal", and we have motivated this focus of our study by the substantial recent work linking pupil diameter to cortical state in the Introduction. Furthermore, we have added a paragraph to the Discussion (3rd paragraph), which reviews the complex evidence pertaining to the link between pupil diameter and specific brainstem systems involved in arousal regulation.

Other examples, like "We hypothesised that decision-makers' uncertainty about their preceding choice shapes serial choice biases by recruiting the arousal systems of the brainstem. These systems are in an ideal position to transform decision uncertainty into adjustments of behaviour: They receive descending projections from frontal brain regions encoding decision uncertainty and have widespread modulatory effects on their cortical target networks." This is a very nice hypothesis but the authors' work has no bearing on it.

We have moved this scenario to the Discussion, where we present it as a possible interpretation of the current findings, which will hopefully motivate future studies to test it. Further, we review the literature on possible neural sources of fluctuations in pupil diameter in the Discussion (3rd paragraph).

Pupil dilation will correlate with numerous brain systems and the origins of the signal or its causality are not studied.

Agreed. The peripheral neural pathways controlling pupil diameter are well characterised, but, indeed, the central brain systems driving this apparatus are not yet fully determined (McDougal and Gamlin, 2008).

That said, we do believe that recent evidence from monkey and human does provide more specific constraints for the interpretation of our data than your statement suggests. Monkey electrophysiology shows that pupil diameter correlates with neural activity in a number of brainstem nuclei involved in attention and arousal, but most strongly (and with shortest latency) in the locus coeruleus (LC) and the superior colliculus (SC) (Joshi et al., 2016; Varazzani et al., 2015; Wang and Munoz, 2015). Ongoing high-resolution fMRI work in humans from our lab covering the entire brainstem converges on similar observations (De Gee JW, Kloosterman NA, Nieuwenhuis S, Knapen T, Donner TH. Decision-related pupil dilation reflects locus coeruleus activity and altered visual evidence accumulation. Program No. 60.21/K30. Neuroscience Meeting Planner. Chicago, IL: Society for Neuroscience, 2015). This work shows that correlations between pupil dilation and fMRI responses are stronger for the LC and SC than for dopaminergic midbrain nuclei, and the basal forebrain. With respect to causality, microstimulation of the LC and SC triggers pupil dilation (Joshi et al., 2016; Wang and Munoz, 2015), again with the shortest latency for the LC.

We now discuss these studies in more detail in the Discussion (3rd paragraph), acknowledging that they paint a complex picture of the central neural drivers of pupil-linked arousal.

"We applied motion energy filtering to the stochastic random dot motion stimuli to compute an estimate of each trial's sensory evidence, which approximates the population response of motion-selective neurons in visual cortical area MT" I think the authors compute the average motion energy which is the not a good measure of what MT neurons encode. Please leave the speculation out of the Results section.

We have re-phrased this statement as follows: *"We applied motion energy filtering to the stochastic random dot motion stimuli, yielding a more fine-grained estimate of the decision-relevant sensory evidence in the stochastic stimuli than the nominal level of motion coherence (Figure 2b,c and Supplementary Methods)."* We have eliminated the reference to area MT.

To clarify the procedure, we computed, separately for each stimulus frame, the directional motion energy along the axis of the nominal motion signal (normalised by subtracting its energy in the opposite direction). Subsequently, we averaged this filter output over time, within each of the two stimulus intervals. We then took the difference between the resulting values for both intervals to compute a scalar measure of the task-relevant sensory evidence to be judged by the observer – the difference in directional motion energy between the two intervals. We have now described this computation more clearly in the Supplementary Methods.

Our rationale for linking our measure of evidence, derived from motion energy filtering, to the neural response of the MT population was as follows. The output of each tuned filter pair is approximately proportional to the spike count (across the whole stimulus interval) of MT neurons with the corresponding direction-selectivity (Britten et al., 1993; Simoncelli and Heeger, 1998). Specifically, the neurons tuned to the direction of the physical stimulus linearly increase their spike count in proportion to the motion coherence, while the neurons tuned to the opposite direction linearly decrease their firing rate, but with a shallower slope (Britten et al., 1993). Consequently, the difference between the responses of these two populations of MT neurons should be proportional to the difference of the outputs of two oppositely oriented filter pairs, which we have calculated. The same should hold, albeit with a smaller proportionality constant, for MT neurons tuned to directions off the task-relevant axis, whereby this proportionality constant would be 0 for neurons tuned in directions orthogonal to the task-relevant axis. Therefore, the difference between the outputs of the two filter pairs should be approximately proportional to the response of the entire population of MT neurons (see also Jazayeri and Movshon, 2006). We acknowledge, however, that this only holds for the MT population response during one of the two stimulus intervals of our experiment. The difference of the population response between these two intervals will, most likely, not be explicitly encoded in the MT response, but rather computed in downstream regions, which maintain a memory of the sensory response during the first interval (e.g. Machens et al., 2005).

"To this end, we analysed reaction time (RT), an established proxy for decision uncertainty." RT is a proxy for just about every type of behavioral measure. The observations are fascinating but I wouldn't call RT an established proxy for uncertainty.

Our reasoning was based on one line of theorising about the link between decision time and uncertainty (Kiani et al., 2014; Fetsch et al., 2014 and the literature reviewed therein). However, this link is debated, and we followed your suggesting to introduce RT in a more neutral, assumption-free

fashion: “In line with previous work (Sanders et al., 2016), RT exhibited all three signatures of decision uncertainty derived in Figure 1 above (Figures 2e and S1b,c) – specifically, decreasing with evidence strength on correct trials but increasing with evidence strength on errors (Figure 2e).” Moreover, we removed the phrase “Having thus established the effectiveness of our manipulation of decision uncertainty, ...”.

There are several other such claims, please remove them from the Introduction and Results section. The Discussion of course the authors are free to speculate about their results might be relevant for neural systems.

As explained above, we have carefully worked through the whole manuscript and we are confident that this issue is now resolved. Thank you for your guidance.

(2) Throughout the paper the authors rely on a single signature of confidence, as shown in Figure 1c. Although I agree that it is the most useful, somewhat counterintuitive signature, I think the authors should try to establish all three known signatures that follow from the signal detection model (Kepecs et al, 2008). Perhaps the most relevant presentation of these are in a recent paper by Sanders et al, 2016. In particular, it would be important to establish how well pupil diameter predicts choice accuracy (Sanders paper Fig 1B) and whether this prediction is beyond what can be made by using decision difficulty on its own (Sanders paper Fig 1D). Importantly, the authors need to show in Figure 2, for RT, that the prediction of accuracy is not below 50% thus showing that it is not "error detection" but confidence. The same applies to the pupillometry data in other Figures. From these two signatures the third might follow but ideally the authors should directly show that the psychometric slope is different for high vs. low pupil diameter.

Thank you for pointing us to this important issue. We now test for all the signatures of uncertainty presented in Sanders et al. (2016). We extended Figure 1 to show all the signatures of the model-derived measure of uncertainty. We have added an additional Supplementary Figure S1 showing that RT exhibited all three signatures. Specifically, we show in this new Supplementary Figure that the prediction of accuracy by RT is not below 50% indicating that RT does not reflect "error detection" but uncertainty. Finally, we also added the corresponding plots for the pupil as new panels in Figure 3 and Figure S3. Also pupil responses, albeit noisier than RT, do not predict accuracies below 50% and are overall more consistent with uncertainty than with error detection.

(3) If I understand the history analysis correctly the authors separately examine stimulus identity and previous choice but do not have a regression on previous outcome. My question is how much does a term in Previous choice X previous outcome affects the regression? Figure 4d does seem to suggest that there is a correlation between choice weight and stimulus weight indicating perhaps an interaction between stimulus and choice.

We now transformed the choice and stimulus weights into weights for repeating a previous correct or incorrect choice (Figure S6c). There was no group-level effect of preferentially repeating/alternating choices after correct vs. error feedback. Similarly, the current Figure 5d suggests that only 7 out of 27 observers fall into a feedback-dependent subspace (in the upper and lower hemi-fields, distant from $y = 0$) of the strategy space, whereas the dominant component of the inter-subject variability is along the x-axis (i.e., between stay or switch).

Relatedly, we also show the pupil- and RT modulation weights for both correct and error trials (Figure S6d). Pupil weights were stronger (and statistically significant) for errors but the basic pattern was similar for both trial outcomes, and there was no statistically significant difference between pupil modulation weights of repeating a previous correct vs. incorrect choice. We now also show the analyses from Figure 4a and 4f separately for correct and error trials in Figure S6a and S6b. Again the patterns are similar for both trial outcomes.

Similarly in Figure 6, the analysis could be done using the previous outcome.

The new Figure S10 shows the correlation between individual choice weights and pupil- and RT-modulation of serial biases after correct and error trials. This revealed a striking dissociation between pupil- and RT-linked effects on serial choice bias: Individual choice tendencies predicted the degree to which pupil modulated serial biases after correct trials, but not after errors, and conversely for RTs. In other words, the RT-dependent bias reduction was most pronounced after incorrect choices, whereas the pupil-dependent alternation boost was most pronounced after correct choices. This result further supports the idea (already alluded to in the original submission) that the pupil- and RT-linked modulatory effects may be mediated by distinct underlying neural systems. Indeed, we believe this new finding opens a very interesting avenue for future investigations into the underlying neural systems (elaborated on in the 2nd last paragraph of the Discussion), which we are currently pursuing.

Thank you again for insightful and constructive comments.

Point-by-point reply to reviewer #2

The manuscript reports a tremendously rigorous and detailed analysis of sequential dependencies in a visual motion discrimination task and their relation to arousal as measured through pupil dilation.

Thank you for these kind words.

The authors demonstrate that post-decision pupil dilation scales with levels of uncertainty inferred from an SDT model of decision uncertainty and from experimentally observed RTs. The size of the pupil response predicted a shift to choice alternation in the perceptual task. The main novelty of the study is linking uncertainty-related arousal to sequential dependencies. The other main finding, that arousal increases with uncertainty and errors, is now established using measures including pupil dilation and heart rate changes: Lempert et al. (2015, PLoS One), Wessel et al. (2011, J Cogn Neurosci), Hajcak et al. (2003, Psychophysiology).

This comment made us realise that we did not explain the conceptual advance offered by the first main result of our study sufficiently clearly. We have added a new Discussion paragraph (2nd paragraph) that elaborates on the relationship between our current findings and the previous work linking human pupil dynamics to uncertainty and performance monitoring. Several important features of our approach allowed us to move beyond this previous work:

- Different from most previous studies (see O'Reilly et al. (2013) for a similar approach), we here unravelled the temporal evolution of uncertainty information in the pupil response, enabling inferences about not only the existence, but also the time course of this information. This is not just a technicality, but key for the interpretation: It is conceptually important to pinpoint whether the uncertainty scaling emerges *during* the decision process, or *after* the decision process, or after the external feedback. By showing that the scaling emerges right at the end of the decision process and before feedback delivery, our analysis establishes a close link between the time course of uncertainty coding in our pupil measurements and in neuronal responses in OFC identified by Kepecs et al. (2008).
- Decision uncertainty, as defined in our framework, may be distinct from subjective confidence ratings: accessing and reporting internal decision uncertainty signals may require additional calibration of these signals, which may introduce substantial differences between decision uncertainty and subjective confidence (Kepecs and Mainen, 2012). For this reason, the observation of uncertainty scaling in our data does not trivially follow from the findings by Lempert et al. (2015). Also note that this study does not show a time a course of the confidence-related component in the response (see previous point).
- The model-based definition of decision uncertainty we used helped dissociate decision uncertainty from error detection, which has previously been linked to pupil dilation (Wessel et al., 2011). In a two-choice task, a signal encoding decision uncertainty should predict behavioural performance over a range from 100% to 50% correct. By contrast, an error detection signal should predict performance over the range 100% to 0% correct (Kepecs and Mainen, 2012; Kepecs et al., 2008). Here, we explicitly tested these predictions, providing clear support for the uncertainty scenario, and against the error detection scenario (Figures S1b and S3c). This crucial difference distinguishes our work from that of Wessel et al. (2011) and Hajcak et al. (2003), who focussed on consciously detected and reported errors, and who did not set out to arbitrate between uncertainty scaling and error detection by quantitatively relating pupil responses to accuracy.
- In our task, decision uncertainty critically depended on *internal* noise (the primary source of the variance in Figure 1a). By contrast, previous studies linking uncertainty to pupil dynamics (de

Berker et al., 2016; Nassar et al., 2012; O'Reilly et al., 2013; Preuschoff et al., 2011) have used tasks in which the primary source of uncertainty was in the observers' environment.

- In contrast to most previous pupillometry studies (de Gee et al., 2014; Lempert et al., 2015; Preuschoff et al., 2011) we comprehensively quantified the predictive effects of pupil-linked arousal on the parameters of choice *beyond* the current trial. Thereby, our work complements recent work on the effects of pupil-linked arousal on learning (Nassar et al., 2012; O'Reilly et al., 2013).

Taken together, our results critically advance the understanding of how internal decision uncertainty is encoded in pupil-linked arousal in humans, in a way that builds a direct bridge to single-unit recording studies of decision uncertainty in animals (Kepecs et al., 2008; Komura et al., 2013; Lak et al., 2014; Teichert et al., 2014).

However, a weakness of the main novel result is that the causality is not established.

We are aware of the limitations of correlative approaches, and we tried to be careful in framing our conclusions. We would like to note, however, that the majority of novel findings in cognitive and systems neuroscience come from correlative studies, with causal approaches often inspired by these previously established correlations.

For example, the seminal modelling and single-unit physiology work by Kepecs et al. (2008), which established a correlation between OFC single-unit activity and decision uncertainty, was followed by a later inactivation study from the same group to establish causality (Lak et al., 2014). Likewise, we are currently probing into the causal effects of specific neuromodulatory systems in guiding uncertainty-dependent behavioural adjustments, using pharmacological interventions (combined with whole-brain MEG recordings) in healthy humans performing the same visual decision-making task. This new study was largely inspired by the (partly predicted and partly unexpected) observations we made in the current study. We hope that our findings will also motivate other groups to use interventional procedures for establishing causal links.

It is not intuitive that increased arousal should result in choice alternation, and the manuscript does not develop a strong case for this prediction (the relevant analysis seems exploratory rather than hypothesis driven).

Thank you for this important comment. It made us realise that the reasoning guiding our analyses was not sufficiently clear in our previous presentation of the findings: While we did not have specific predictions about the type of arousal-dependent adjustment(s) of subsequent choice behaviour, we did strongly predict that *some* computational element of the decision process should change. This more general prediction is firmly grounded in recent animal physiology work establishing strong and widespread, pupil-dependent changes in the dynamics of cortical circuits (McGinley et al., 2015). Because cortical circuits contribute to decision-making (Gold and Shadlen, 2007), any phasic change in pupil-linked arousal state should manifest itself in at least one of the main components of perceptual decisions: response time, perceptual sensitivity, lapse rate, or bias (further partitioned into overall and serial bias).

We systematically tested the effects of previous pupil responses on these components of the decision and found a, so far unknown, specific effect on serial bias. Indeed, this result could not have been predicted from the previous literature. In our mind, however, the surprise about this

finding is not a weakness, but rather a strength of the current study: The fact that our results, at least in part, challenge current theories of the impact of arousal on decision-making underscores the significance of the results. Also, please note that we did not just stop at that basic observation (Figure 4), but used a novel modelling approach to try and unravel the underlying process (Figure 5).

That said, our findings are remarkably well in line with recent rodent work implicating the LC-NE system in the serial correlation structure of choice behaviour (Tervo et al., 2014). Future work, which we and other labs are now performing, will be needed to pinpoint the neural mechanisms underlying these novel modulatory effects on the serial correlation structure of choice behaviour.

We have made substantial changes in the presentation of our results in light of your comment. The goal of these was to clarify the reasoning underlying, and development of, our analyses. In the revised Introduction, we have now explicated that we have systematically quantified the effect of arousal on all major components of decision-making. In the new Figure 4, we now show the effect of pupil responses and RT on all main components of the decision: perceptual sensitivity, lapse rate, response times, absolute choice bias, and serial choice bias. Pupil responses and RT significantly predicted a modulation of serial choice bias, but not a change in absolute bias, signed bias or perceptual sensitivity, and had only a weak (and non-significant) effect on lapse rate and post-error slowing. This pattern of results motivated our more detailed analysis of serial choice bias shown in Figure 5. Correspondingly, we have also re-organized, and largely re-written, the part of the Results section dealing with the predictive effects of pupil responses (see sections “Pupil-linked arousal alters subsequent choice behaviour” and “Pupil-linked arousal predicts choice alternation”).

Equally, there are many potential routes by which frontally-coded uncertainty could affect choice, some of which seem better established or more plausible than the one proposed. Why not a direct effect within frontal cortex (Botvinick et al., 2001, Psych Rev), or between frontal cortex and basal ganglia?

We agree. Indeed, parts of frontal cortex that encode decision conflict and task utility, also correlate with pupil responses (Ebitz and Platt, 2015) and may well play an important role in modulating serial biases. That said, the observed dissociation between pupil- and RT-linked effects on serial choice bias suggests that decision uncertainty signals are propagated along distinct neural pathways, one linked to pupil responses and the other to RT, which then shape serial choice biases in different ways: Pupil-linked arousal boosted choice alternation per se, rather than eliminating each observer’s intrinsic serial bias; RT, on the other hand, predicted a reduction in individual serial bias, regardless of whether this bias is towards repetition or alternation (Figure 5). (See also the reply to your point below regarding the lack of correlation between pupil responses and RT.) A new group analysis using the pupil- and RT modulation weights after error and correct trials revealed another striking difference between the two (Figure S10).

We now elaborate on this possible dissociation of pupil- and RT-linked effects at the neural systems level in the Discussion section (2nd last paragraph):

“The dissociation between pupil- and RT-linked modulatory effects (Figure 5 and S10) on serial choice bias suggests that decision uncertainty signals were propagated along distinct central neural pathways, one linked to pupil responses and the other to RT, which then shaped serial choice biases in different ways. Even if the same uncertainty signals fed into these pathways, they might have become decoupled through independent internal noise. Specifically, it is tempting to speculate that the pupil-linked alternation boost reflected neuromodulator release from brainstem centres (such as noradrenaline from the LC (Tervo et al., 2014)), whereas RT-linked bias reduction was driven by

frontal cortical areas involved in explicit performance monitoring and top-down control (such as anterior cingulate cortex) (Botvinick et al., 2001; Ebitz and Platt, 2015; Yeung et al., 2004). Top-down effects of prefrontal cortex on decision-making (Botvinick et al., 2001; Miller and Cohen, 2001) are commonly associated with explicit strategic effects that are adaptive within the experimental task. Indeed, the RT-linked modulation of serial bias was adaptive, in that it generally reduced observers' intrinsic serial bias. By contrast, pupil-linked arousal modulated serial choice patterns in a way that was maladaptive for part of the observers (the alternators). This finding might be related to the observation that maladaptive serial choice biases remain prevalent even in highly trained observers who know the statistics of the task (Fernberger, 1920; Fründ et al., 2014). Taken together, the dissociation between pupil- and RT-linked effects suggest that serial choice biases result from a complex interplay between low-level, pupil-linked arousal systems and higher-level systems for strategic control. Future studies should pinpoint the neural systems underlying these distinct effects, as well as their interactions (Tervo et al., 2014)."

One recent suggestion is that confidence can guide perceptual learning by augmenting well-known reinforcement learning processes (Guggenmos et al., 2016, eLife), which could produce the effects reported here.

This is an interesting suggestion, and we have read the above study with great interest. However, although both processes, the learning process in the above study and pupil responses in ours, were driven by decision uncertainty/confidence, two aspects of the data suggest that they are nonetheless distinct. First, perceptual learning is (as in the above study) commonly measured as improvement in perceptual sensitivity – which did not depend on pupil or RT in our study (Figure 4c,h). Second, the learning process in the above study (as well as in the related studies by Kahnt et al., 2011; and Law and Gold, 2008, 2009) operates on a slow timescale, across sessions. By contrast, as we now show in the new Figure S2, the patterns of sequential effects remain rather stable over sessions. Hence, the pupil- and RT-linked processes modulate decision-making on a shorter, more local timescale.

That is not to say that we dismiss any link between reinforcement learning and the modulation of bias we have pinpointed here. In fact, we believe this relationship is a very interesting avenue for future study (e.g. Yu and Cohen, 2008). This is only to say that we feel uncomfortable drawing strong connections between the above and our current study in light of the available evidence.

That said, I found the treatment of the RT/uncertainty/dilation relationship confusing in places. The RT analysis in Figure 1 is first interpreted to establish the effectiveness of the manipulation of decision uncertainty, but in the next section the weak relationship between RT and pupil responses is also interpreted positively. Could the authors clarify their thinking about the relationship between RT, uncertainty and pupil responses?

Thank you for pointing us to this issue, which was indeed confusing. We have now resolved this by presenting the RT scaling as an independent finding (now presented in Figure S1). Indeed, we had no strong prior on the scaling of RT with decision uncertainty, other than that some earlier studies have reported RT to scale with decision uncertainty (Kiani et al., 2014; Sanders et al., 2016). Our RT results are consistent with these observations. We also removed the phrase “*Having thus established the effectiveness of our manipulation of decision uncertainty (...)*”.

Our interpretation of the weak correlation between RT and pupil dilation is as follows. The dissociation between pupil- and RT-linked effects on serial choice bias suggests that internal uncertainty signals are propagated along distinct neural pathways, one linked to pupil responses and the other to RT, which then shape serial choice biases in different ways. Even if the same uncertainty signals feed into these pathways, they might become largely decoupled due to independent (and strong) internal noise. Consistent with these ideas, the trial-by-trial correlation between the pre-feedback pupil responses and RT was negligible, and the correlations of each of these measures with uncertainty were, albeit highly significant, also imperfect. We have added a new paragraph to the Discussion (2nd last paragraph, quoted above) to clarify this thinking.

Tangentially related, to what extent is the difference between pupil response's and RT's relationship with choice weight an artifact of removing log-transformed RT from the trial-to-trial fluctuations of pre-feedback pupil responses via linear regression?

The result in Figure 5e did not depend on the removal of RT from pupil responses (and vice versa): The figure below shows qualitatively identical effects as Figure 5, but without removing the shared variance between the pupil and RT regressors. For this analysis, we used two separate regression models; one where only pupil values modulated serial effects (left), and another one where only log-transformed RT modulated serial effects (right). This result is now reported in the Results text (section “Pupil-linked arousal predicts choice alternation”, 1st paragraph).

We think that removing fluctuations in RT from the pupil signal is the preferred approach. The rationale for doing so is as follows: Because of the temporal low-pass characteristics of the peripheral pupil apparatus (de Gee et al, 2014; Korn et al, 2016), the amplitude of pupil responses scales with the duration of neural input (even if that input has constant amplitude). Consequently, longer RTs might translate into bigger pupil response amplitudes simply because of accumulation of the driving neural signal. This is a conceptually uninteresting source of possible correlation between pupil responses and RT, apart from the (conceptually interesting) possibility that trial-to-trial fluctuations of both variables might be driven by a common central uncertainty signal. To eliminate this ambiguity, we opted for showing the analysis with all shared variance removed in the main paper. Regardless of the theoretical motivation, the low overall correlation between pupil and RT shows that this issue was of little practical relevance in the current study.

One aspect that I did not see discussed is the extent to which strength of serial choice biases varied with the difficulty of the current trial. Presumably biases were stronger on more difficult trials, where choices would be less dominated by sensory evidence?

True. We now include the relative contribution of history effects in a new panel of Figure 5 (b), where this shows very clearly.

Partly related to this point, p.14 discusses effects of small serial correlations between stimuli. Did the authors also rule out possible effects of serial correlations between difficulty levels? From the description it re and vice versa. Could this dependency, if present, contribute to the apparent effects of uncertainty on serial choice biases, as measured either through pupil responses or RT?

We tested the correlation between subsequent difficulty levels for each session, and report this in the Results (section “Pupil-linked arousal predicts choice alternation”, 3rd paragraph). Across observers, we found a very small autocorrelation (average $r = -0.016$, range -0.061 to 0.028). This autocorrelation was not related to observers’ individual choice weights ($r = -0.119$, $p = 0.553$, $Bf_{10} = 0.177$) or stimulus weights ($r = -0.139$, $p = 0.488$, $Bf_{10} = 0.189$). We also did not find these autocorrelations in evidence strength to be related to pupil x choice weights ($r = -0.029$, $p = 0.886$, $Bf_{10} = 0.150$) or RT x choice weights ($r = -0.038$, $p = 0.851$, $Bf_{10} = 0.151$).

Thank you again for insightful and constructive comments.

Point-by-point reply to reviewer #3

Overall I see the results as timely, interesting and generally well communicated.

Thank you for these kind words.

The authors provide a very concise narrative to describe the key findings, but in places I think it would have been useful to have some more basic analysis. In particular, choice or accuracy data would have been useful in evaluating the evidence strength metrics provided in Figure 2.

Thank you for this helpful suggestion. We have now included psychometric and chronometric functions in Figure 2d, showing the effects of evidence strength on accuracy and RT.

In addition, since the data were collected across multiple sessions, it would be good to know how performance (accuracy & RT) change within and across sessions and to ensure that the key findings are not simply due to pupil and behavior both changing as a function of time/training. The authors should also include some analysis of how choice history curves (Figure 4) change over training. Given that history was not predictive of stimulus, I would expect that these effects shrink over time. Any such effects should be accounted for in the pupil and RT analyses of sequential bias (Figure 5).

Because learning was not the scope of the current study, we focused our analyses on quantifying behaviour in the steady state in which psychophysical thresholds had stabilized. Consequently, we had observers do an initial practice session consisting of general instruction and 500 trials. During this session, we observed strong improvements in performance (differing substantially between observers), and we changed the sets of presented motion coherence levels contingent on observers' performance. The data from these practice sessions were not used in the analyses presented in the manuscript. This aspect of our experimental design is now clarified in the Methods section.

We added Figure S2a showing, per session of 500 trials, psychometric and chronometric functions (as in Figure 2d). We now also show per-session choice kernels, both at the group (Figure S2b) and individual (Figure S2c) level. Overall, two observations stand out: (1) psychometric functions (including history effects) remained relatively stable over the course of the experiment; (2) RTs became faster over sessions.

Regarding observation (1), we computed repetition probability for three bins of pupil responses, separately in each of the five sessions. Repeated measures ANOVA revealed neither a main effect of session ($F_{(4,104)} = 1.658$, $p = 0.165$, $Bf_{10} = 0.082$) nor an interaction between session and pupil bin ($F_{(8,208)} = 1.281$, $p = 0.255$, $Bf_{10} = 0.010$) on repetition probability (reported in the caption of Figure S2). Since psychometric functions and history kernels were stable over sessions, we opted not to explicitly account for session-by-session changes in our model-based analyses of serial bias. This would have only made our (already complex) regression model of the modulation of serial bias even more complex, with little motivation from the data. Regarding observation (2), the decrease of RT made us realise that between-session differences in RT may have confounded our analyses of RT-linked modulation of serial biases. We now account for these across-session differences by z-scoring log-transformed RTs, within each block of 50 trials. This puts the normalisation RTs on equal footing with that of pupil responses (which were likewise z-scored per block). Notably, this change only made the effects of RT on reducing serial bias, both in the model-free (Figure 4f) and model-based analyses (Figure 5e), more robust.

It would be useful if the authors could provide example RT distributions and comment on the level of overall variability in RTs.

Thank you for this useful suggestion. We added RT distributions in S1a. These show RT distributions for all trials combined, for five levels of evidence strength, and separately for error and correct responses. Furthermore, the figure below shows the cumulative distribution of RTs from stimulus offset (mean \pm s.e.m. across observers), which indicates clearly that the vast majority of RTs was confined to a comparably narrow range from about 100 to 750 ms after stimulus offset (the go-signal for the choice report) – as expected for an interrogation protocol in which the stimulus viewing time is determined by the experiment and not by the decision-maker.

In particular, it was not clear to me whether the reduced sequential dependencies for long RT trials could simply be due to the additional passage of time.

We tested for such an effect in two approaches. Both indicated that the reduced passage of time could not explain the reduction of serial bias for longer RTs. The results are now shown in the new Figure S8a and S8b, along with another result on interval timing (Figure S8c).

In general, we quantified the passage of time between trials in terms of the latencies between the onset of each test stimulus and the onset of the next trial's test stimulus. In the first approach, we then removed (using linear regression) these trial-by-trial latencies from the vector of normalised RTs. RTs correlated with the time between stimuli (mean Spearman's rho: 0.311, range 0.078 to 0.738). Yet, removing these trial-by-trial latencies did not abolish the effect of RTs on serial choice bias (Figure S8a, $F_{(2,52)} = 9.236$, $p < 0.001$, $Bf_{10} = 97.298$).

In the second approach, we repeated the analysis from Figure 4, but now binning by the above trial-by-trial latencies between test stimuli. We found no effect of these latencies on repetition probability (Figure S8b, $F_{(2,52)} = 0.817$, $p = 0.447$, $Bf_{10} = 0.202$).

The authors control for RT in assessing the effects of pupil diameter in sequential bias, however, as far as I can tell, they did not control for previous trial accuracy, which explains a large proportion of the variance in the pupil data. It would be good to know whether trial accuracy itself affects sequential bias and to ensure that the effects of pupil diameter are not completely mediated by accuracy differences.

To confirm that the effect of pupil responses on serial bias is not solely due to differences between pupil responses on correct and error trials, we repeated the analyses shown in the current Figure 4 after splitting by both pupil response and trial outcome (Figure S6a). We found that the decrease in choice repetition after high pupil responses was similar for correct and error trials, and statistically significant when tested on error trials only (it was marginally significant at $p = 0.072$ when tested on

correct trials only). Taken together, we found no evidence for the effect of pupil responses on serial bias was being mediated by accuracy differences.

One lingering question in my mind is whether the relationship between alternation-type sequential dependencies and pupil diameter occurs over a short timescale (e.g. trial-to-trial) or whether they go through slow and synchronized fluctuations. To this effect it would be useful if the authors could conduct their analyses (e.g. Fig 6 B) on time shifted data to, for example, determine whether the pre-feedback pupil response on trial t has any relationship to the sequential biases observed for trials t or $t+2$.

In Figure S7b, we now show the pupil x choice weights as a function of lag (up to 7 past trials). There is no evidence for an effect of pupil responses on serial bias beyond 1 trial in the past.

Page 10, paragraph 2, the authors say that: "the bias component of choices became less history-dependent, more neutral and thereby better adapted to the statistics of the task", but later on, they look more closely and show that in reality, this was only true for some subjects. In particular, the subjects that showed alternation biases were more biased on trials following a large pupil response. It would be interesting to know whether this basic idea bears out by looking at how accuracy relates to pupil diameter for perseverative versus alternating subjects.

We found no effect of pupil responses on subsequent trial accuracy, within either subgroup of observers (see left panel below). As an alternative, potentially more sensitive, measure of overall task performance, we also fit cumulative Weibull functions to accuracy as a function of sensory evidence, after binning by previous trial pupil response. The results are shown in the right panel below. Again, there was no clear pupil modulation effect, within either subgroup of observers.

The absence of a pupil effect on accuracy/sensitivity is likely due to the nature of our task, in which accuracy is largely determined by large trial-to-trial variations in the strength of sensory evidence (Figure 2d). A design with constant evidence strength, titrated to psychophysical threshold, might be more sensitive for detecting changes in accuracy as a function of previous trial pupil response.

The authors discount the idea that conditional (error/correct) changes in RT over time could be due to task strategy "since the stimulus duration was fixed, observers could not accumulate more evidence by slowing down decisions on uncertain trials." However, I am not totally convinced by this. Couldn't subjects make a decision during motion viewing on some trials leading to action preparation during the 500 ms viewing interval and faster RTs?

Your comment made us realise that our previous statement was ambiguous. Rather than claiming that observers always integrated all available evidence (which our current data does not allow), we wanted to emphasise that, due to the nature of the task, observers did not control the amount of evidence available to them. We have rephrased this sentence to “*This was true despite the interrogation protocol, in which the test stimulus had a fixed duration, its offset prompted the choice, and observers were instructed to maximise accuracy without speed pressure (response deadline was 3 seconds after test offset).*” We hope you agree that this now correctly emphasises the task structure, without making unwarranted assumptions about observers’ strategies.

Once again, knowing what the RT distributions looked like might be informative in this regard.

We now show RT distributions in Figure S1a. Across all stimuli, median RT was 360 ms (range 150 – 710 ms across subjects, Figure S1a, left). For only those stimuli with the strongest evidence, median RT from stimulus offset was 280 ms (range 100 – 670 ms across subjects, Figure S1a, middle). Given that typical response times in simple cue detection tasks are around 200 ms (Hick, 1952; Luce, 1991), it seems unlikely (at least on average) that observers had their choice made and motor plan prepared already before stimulus offset. However, again, we do not wish make strong claims about the strategy used by observers: clearly, some decisions might be completed already within the second stimulus interval, despite our use of an interrogation protocol (possibly using a process such as bounded diffusion (Kiani et al., 2008)).

The authors go on to discard the notion that mental effort could explain the linkage between pre-feedback pupil diameter and evidence (conditioned on accuracy). However, along the lines of my previous comment, you could imagine operationalizing mental effort as something like: "time accumulating information". With such a definition, and with an accumulation process incorporating bound separation variability and truncated by the stimulus offset, I could imagine that you might get a similar predicted pattern. This is particularly relevant giving that DDM modeling of value-based decision task behavior has suggested that arousal relates to an increased decision threshold, which would lead to a longer accumulation time⁵. Not at all to say that this detracts from the authors primary claim regarding the downstream effects of this signal on subsequent behavior, but I think that if the authors would really like to rule out mental effort they should consider more robust versions of it.

Both you and reviewer #4 commented on our discussion of effort. From these reactions, we realised that this discussion was not helpful, precisely because of a lack of a commonly agreed operational definition of the construct “effort”. This term can be operationally defined in different ways. In our original submission, we aimed to argue with one common definition of effort, but we acknowledge that there alternatives, which might lead to changes in interpretation. We think this makes it difficult to evaluate whether or not effort could explain our results, which casts doubt about the utility of the construct (see also reviewer #4).

Specifically, we feel uncomfortable discussing evidence accumulation in the context of the current study, because there is nothing in our measurements that allows us to make inferences about things like drift and decision threshold; this would have required a different design, measurements, and/or analyses. To avoid over-interpretation, we prefer to stick to the quantities we can robustly estimate.

So, rather than elaborating on a range of possible definitions of effort, and speculating about our data, we opted to focus on our (precise) definition of decision uncertainty and drop our discussion of effort.

The authors claim that the effect of pupil response on subsequent behavior was specific for pre-feedback pupil responses based on a failure to find an effect in the post-feedback responses after regressing out shared variance, however it seems that supporting this claim would require the converse to be true. Otherwise the pupil measures may just be too highly correlated to see any effects in the unique variance.

We realised that our choice of the term “specific” misleadingly overstated our claims: We simply aimed to convey the idea that the information driving pupil-linked modulation of serial effects was already present *before* the external presentation of feedback; the post-feedback pupil signal did not add any predictive information.

To clarify this point, we now present this result in the following way in the Results section “Pupil-linked arousal predicts choice alternation”, 5th paragraph: *“The pupil response after feedback did not contain information predictive of serial choice bias, beyond the information already present during the pre-feedback interval. The post-feedback pupil responses similarly predicted modulation of serial choice biases, but no longer did so when removing (via linear regression) variance explained by pre-feedback pupil responses from the post-feedback pupil signal (see Figure S9).”*

It would be useful if the authors could provide more discussion of the linkage between the effects of arousal on sequential biases and relevant ideas regarding the role of arousal in learning and decision-making. (...) That said, the specifics of the findings may not fit exactly, as neither of those theories would account for the promotion of alternation. In any case, for the results of this study to be interpreted broadly, it would be useful for them to be cast in the context of existing theory.

Thank you for raising this important point: We now realise that our Discussion was missing a clear link to existing theory – and we agree that there are interesting overlaps with, as well as important differences to, influential theoretical accounts. We had discussed our results in the light of these different accounts at length internally, but now see that the previous version of the paper addressed them in an insufficient manner.

We have added two new paragraphs to the Discussion section (6th and 7th paragraph). The first of those relates our findings to proposed theories on the functional role of phasic arousal responses. Specifically, we emphasise that individual differences paint a more complex picture than the existing accounts, which challenges these theories and opens up new avenues for future study. The second paragraph focuses on the temporal dynamics of our signal with respect to previous accounts of tonic vs. phasic arousal responses, also distinguishing our results from previous work.

Does the evidence conditioned on choice relationship to pupil diameter hold until the beginning of the subsequent trial? If so, is pupil diameter at the beginning of a trial predictive of the sequential bias that will be displayed on that trial?

We found a much weaker, but qualitatively similar pattern when splitting baseline pupil diameter by previous trial evidence strength and trial accuracy, as shown in the figure below.

Likewise, baseline pupil diameter was also not significantly related to the current trial's serial bias (Figure S7a). Relatedly, pre-feedback pupil responses did not predict the modulation of serial choice bias beyond one trial (Figure S7b). Taken together, these results raise the interesting possibility that pupil-linked arousal only affects subsequent choice behaviour through the interaction with cortical circuitry that is active around the time of the choice.

Page 3, paragraph 2: "this quantity is encoded..."

This sentence could be read to mean that uncertainty is represented in one set of regions in monkeys and a different set of regions in rodents.

We have changed this sentence to: "... findings from animal physiology showing that neurons in a number of brain regions encode decision uncertainty, as defined in Figure 1."

Thank you again for insightful and constructive comments.

Point-by-point reply to reviewer #4

The paper is well-written, and the methods are thorough and the work follows their own previous work that has contributed to establishing pupil responses as an important physiological readout that provides a window into the neural dynamics steering cognition. The paper itself contains a considerable amount of work, and the results appear sound, with several control analyses presented.

Thank you for these kind words.

The paper is a nice extension of previous work, and aligns with reports that link changes in pupil diameter with uncertainty computations, and these in turn with behavioural changes. At the same time, it would be helpful to see the present work more directly discussed with other recent efforts to link uncertainty with pupil dynamics - how does the present work extend these other reports? (e.g. Preuschoff et al 2011; Nassar et al 2012; de Berker et al 2016; de Gee et al 2014).

Thank you for this very helpful suggestion. In the Discussion (2nd paragraph), we now include a new paragraph clarifying in detail how our approach and findings extend these previous reports.

A related point relates to the specific type of decision uncertainty the authors here investigate and quantify. How does this formally relate to other forms of uncertainty, as addressed in some of the papers mentioned above? This point seems relevant not just for relating it to other work, but also the issue of decision uncertainty, response conflict, and reward expectancy in the present study highly overlapping. Put simply, if these three quantities cannot be divorced, then is the construct validity of the paper limited? That is, if one agrees that these three quantities are related, the perhaps it would be useful to recognise earlier on in the paper that decision uncertainty is one of the possible variables that pupil diameter can reflect.

We now realise that our discussion of response conflict, and reward expectancy was not useful. There may, or may not, be partial overlap between decision uncertainty and these two variables. Yet, it is not easy, and beyond the scope of this paper, to formally distinguish between these quantities. Rather than considering verbal descriptions of these different concepts, we opted to focus on one concept that is theoretically principled, defined in a mathematically explicit and parsimonious way, well-suited for our task, and grounded in single-cell physiology work, which motivated our study (Kepecs et al., 2008; Komura et al., 2013; Lak et al., 2014; Teichert et al., 2014): decision uncertainty. We have no difficulty with readers concluding that the results could also be interpreted in terms of response conflict or reward expectancy.

For the sake of parsimony and theoretical clarity, we chose to drop the discussion of response conflict and reward expectancy from the Discussion. Instead, we now more broadly discuss a number of previous studies linking pupil dynamics to various forms uncertainty and relate our findings to this existing literature.

What is the optimal behaviour in the present task? If I understand correctly, each stimulus category was followed by itself or its opposite alternative with equal probability.

From the task description I wasn't sure whether there was any change in stochasticity to learn and track from? If not, why would changes in learning rates be of use here?

We grouped these comments together, and would like to reply to them jointly. Both made us realise that we had insufficiently pointed out the maladaptive nature of serial choice biases in our task.

The optimal policy in our task (as most common in near-threshold psychophysics) is to ignore stimulus history and make choices based only on the current trial's evidence. Even if maladaptive, serial biases are well-known since a long time to robustly occur in tasks of this kind, in many trained observers (Fründ et al., 2014).

We now include the following statement in the Results section "Pupil-linked arousal alters subsequent choice behaviour", 1st paragraph: *"Because in our task (as common in laboratory choice tasks), the sensory evidence was independent across trials, any serial bias was maladaptive, reducing observers' performance below the optimum they could achieve given their perceptual sensitivity."*

Why should we then expect a tendency for choice alteration under high decision uncertainty, and would this be in any way adaptive here? I guess this could be the case simply because there was a tendency for predominant choice bias across individuals, i.e. to repeat choices. If the case, perhaps the authors can explain this in more detail.

Correct – the increase in choice alternation after high pupil responses or RT is adaptive for those individuals who have a tendency to repeat their choices. Because these individuals were in the majority in our sample, the increase in choice alternation was adaptive at the group-level (on average).

However, as we show with our analysis of the individual differences, this alternation boost effect was not adaptive for each individual: it was maladaptive for those observers who already tended to alternate.

Related, there is a long-standing literature on choice biases, recency effects and post-error slowing in the psychological literature. It would be good to be reassured how the present work relates to this earlier work.

While serial choice biases in decision-making *per se* have been firmly established a long time ago, the finding that they are dynamically *modulated* by pupil-linked arousal is entirely novel. In fact, to the best of our knowledge, such a modulatory role has not yet been established for any other physiological signal. This is the most important way in which the present work differs from the existing work on serial choice bias.

In general, there exist two, largely disjoint, bodies of literature on sequential effects in decision-making. The first is on speeded RT tasks, in which serial effects are usually interpreted in terms of speed-accuracy trade-off and modelled by means of sequential sampling models (e.g., as automatic variations in the starting point, or as strategic top-down effects on subsequent decisions, for example decision bounds (Gao et al., 2009)).

The second literature focuses on near-threshold psychophysical tasks with interrogation protocol – in other words, without speed pressure but with systematic variations of stimulus strength over a range spanning psychophysical threshold. Here, psychometric function modelling is commonly used for quantifying selective serial biases (Abrahamyan et al., 2016; Busse et al., 2011; Fründ et al., 2014). Our current design is of this kind, and thus lends itself very naturally to extending this latter type of approach, as a solid platform for quantifying the pupil-dependent modulation of serial bias.

There exists only one study (now cited in our paper), which has proposed a similar modulatory role of pupil-linked arousal on another form of sequential effect in decision-making, post-error slowing in speeded RT tasks (Murphy et al., 2016) (see also next point below). Our findings clearly differ from this study in terms of the type of sequential effect that is modulated.

Generally speaking, the surprising, and novel, aspect of our finding is that pupil-linked arousal can also have more specific effects that are selective for a particular choice (as opposed to an overall slowing-down of responses). This specificity challenges current thinking about the functional impact of arousal systems, more so than previous reports linking pupil responses to post-error slowing.

Also related, did the authors explicitly account for post-error slowing, and how is this related to pupil dilatation? Unless addressed in earlier work that I am not aware of, this could be a useful contribution to these lines of work.

Thank you for pointing this out. We added post-error slowing to the current Figure 4e and 4j. The data show a trend, but non-significant towards increased post-error slowing after high pupil responses, in line with previous work (Murphy et al., 2016). The lack of significance of this effect is not unexpected, and we do not want to make a strong point of this null finding: Our design is simply not well suited for measuring post-error slowing, because it did not explicitly reward speeded responses, and large variations in evidence strength dominated fluctuations in RT.

I may have missed this, but what was the % of correct/incorrect trials? Did performance relate to their individual tendency to be a choice alternator, or a repeater?

Figure 2d now shows the percentage of correct responses (as well as RTs) as a function of evidence strength. We found no correlation between individual perceptual thresholds and the tendency to repeat or alternate choices (mean $r = -0.001$, $p = 0.996$).

If I understand the task instruction correctly, then I think the term "reaction time" should be replaced by "response time", as there was no premium on responding quickly.

We have now changed this throughout.

In that case, some of the discussion, especially when relating to drift models and decision boundaries, may require revision. Similarly, it is then not surprising that no SAT was observed.

Given the nature of our task, it is not entirely unexpected that we did not find evidence for SAT. We have now removed explicit mention of drift diffusion models and decision boundaries from the

Discussion. We believe that the signal detection theory framework presented here provides the most parsimonious account of our results from the current task, without relying on the details of sequential sampling models.

On page 10, the authors state that post-feedback pupil responses predicted modulation of serial choice biases, but no longer did so when removing via linear regression variance explained by pre-feedback pupil responses from the post-feedback pupil signal. It may be worth pre-empting this analysis on page 6.

Thank you for this suggestion, which we considered during the revision of the manuscript. Given the current flow of the paper, however, we found that previewing this result before presenting our analysis of serial choice bias would break the flow of the Results section. We thus chose to present this specific statement, about the timing of predictive information in the pupil signal, shortly after having described the rationale and findings on serial choice biases.

If I understand correctly, 8 participants did not show a tendency for choice alteration, and these were omitted from some analyses. But some clarification is warranted how and when these data were used - for example, if choice biases were not advantageous in the present task, then it seems more difficult to justify to exclude these participants, as would be, for example, if this was a learning experiment and these participants had simply not learned anything.

We apologise for any confusion on this issue, which (we assume) stems from the fact that we have reported the numbers of repeaters and alternators, for whom the choice weights were significantly different from zero. However, we included *all* 27 participants in *all* analyses presented – including those analyses, in which we split the group into subgroups of repeaters and alternators (Figure 5f and 5g).

To avoid confusion, we now clarified this in the Results and Methods section. For example, we added the sentence “*We included all observers in each analyses presented in the paper.*” to the Methods section “Participants and sample size”.

Am I correct that the 7-back analysis is motivated by the previous paper by Fründ? How does that analysis relate to the use of a "decay" function, in which a single parameter could determine how many past trials influence the current choice process?

Yes, the 7-back analysis was proposed by Fründ et al. (2014). The results of this analysis can be related to a (e.g., decaying) function that describes the influence of previous events on current choices. However, the analysis does not impose a particular functional form (e.g. exponential), but rather estimates the contribution of each of the past seven trials. Not for all observers do the results follow a simple monotonic decay; some show more complex, non-monotonic patterns.

Would the authors expect any pupil responses to auditory beeps, as suggested in earlier work? I wonder if this should be accounted for, and would take the author's guidance on this point. Related, would one expect a pupil response to the button press, and if so, is carry-over a possible concern?

Page 16: Do I understand correctly that the pupil signal was in essence quantified as the peak value at 250ms before feedback delivery? Can the authors explain how this account for any overlapping between different quantities that may influence pupil, e.g. beeps, uncertainty, stimulus offset etc? I may have missed this from the methods, in which case I would kindly ask the authors to point me to the respective sections.

Again, we grouped your two comments above together, and would like to reply to them jointly.

Yes, we defined our single-trial measure of pre-feedback pupil response as the mean pupil diameter in the 250 ms before feedback delivery. And: Yes, one would indeed expect pupil responses to beeps, visual transients and button press, which we observe (Figure 3a). However, this is of no concern for the conclusions of this paper, for the following reasons.

First, the scaling of pupil responses with evidence strength, shown in Figure 3, cannot be explained by pupil responses to beeps and stimulus on- or offset. The beta weights in Figure 3b estimated the impact of trial-by-trial variations in evidence strength on pupil diameter fluctuations *around* the mean pupil response shown in Figure 3a (this mean is captured by the intercept of the regression model). For one thing, stimulus on- or offset, and beeps were constant across trials. For another, the random variations of *the intervals between* these task events (i.e., from s1 to s2, and from button press to feedback delivery, Figure 2a) were orthogonal to the trial-to-trial variations in sensory evidence. Consequently, neither the presence of the beeps, stimuli, or motor responses, nor their variable timing can account for the significant deviation of the beta weights in Figure 3 from 0; only variations in evidence strength can account for that.

Second, the predictive effect of pupil responses on serial choice bias also cannot be explained by pupil responses to beeps and stimulus on- or offset. The variable trial intervals between these external events may have indeed contributed to trial-to-trial variations in pupil responses (due to linear superposition of sluggish pupil responses to successive transient neural inputs). However, the random jitter in these intervals makes it unlikely that this would translate into any systematic relationship to subsequent behaviour (unless passage of time play a role, which it does not, see our reply to reviewer #3).

To fully address your comment and pre-empt the above possible confound, we added a new control analysis shown in Figure S8c (and described in the Results). We removed these trial-to-trial interval durations from the vector of pupil responses using linear regression. We then repeated the analysis shown in Figure 5e. Pupil responses were only weakly correlated to the interval between s1 and s2 (mean Spearman's rho -0.007, range -0.052 to 0.044, significant in 3 out of 27 observers) and the interval between button press and feedback (mean Spearman's rho 0.053, range -0.034 to 0.292, significant in 13 out of 27 observers). Removing this variance from trial-by-trial pupil responses did not change the predictive effect of pupil responses on serial choice bias.

Could the authors report in more detail what the results of their artefact removal procedure was? For example, previous work has reported the occurrence of subjects with blinks etc. requiring interpolation of >50%, which may warrant exclusion of those subjects (e.g. Browning et al 2015).

Thank you for pointing this out. We now more clearly state the rationale for our pre-processing choices the Methods section “Pupillometry”. Briefly, we did not reject any trials (nor entire observers) due to bad pupil data: Rather, we linearly interpolated periods of blinks or missing data. Across the group, only 7.7% (std 13.3%) of trials contained more than 50% samples that were interpolated. This rather low percentage is likely due to our observers being highly practiced in the task (from the training session) when we recorded data during the main experimental sessions.

We included all trials from all five main sessions (i.e., excluding the practice session) in the analyses reported in this paper because this was important for modelling of pupil-linked modulation of serial choice biases. The time series of consecutive trial-wise stimuli, choices, RTs and pupil responses was necessary for the regression model. We now state this rationale and the number of interpolated trials in the Supplementary Methods.

Intro: "this quantity is encoded..." This sentence reads as if those regions are established as encoding decision uncertainty, with no other candidates. Is that so?

No, the regions mentioned are merely the ones where uncertainty encoding has been established. We have now rephrased this statement as follows: *“This, in turn, bridges to findings from animal physiology showing that neurons in a number of brain regions encode decision uncertainty, as defined in Figure 1 (Kepecs et al., 2008; Komura et al., 2013; Lak et al., 2014; Teichert et al., 2014).”*

On page 6, the last paragraph seems to engage in a strawman argument. Given quite a large number of pupil studies over the past years, would anyone still argue that pupil dilatation relates to "effort"? Besides, what does effort then actually mean, it seems a poorly operationalized term, perhaps a heuristic at best.

Thank you for this comment, which, together with a comment made by reviewer #3, made us realise that our discussion of effort was not helpful. We agree with you that effort is a construct with many possible operational definitions: In the previous version, we tried to argue against one common definition of effort, but it is now clear to us that this was confusing. Consequently, we have now removed the paragraph on effort from the paper.

Thank you again for insightful and constructive comments.

References

- Abrahamyan, A., Silva, L.L., Dakin, S.C., Carandini, M., and Gardner, J.L. (2016). Adaptable history biases in human perceptual decisions. *Proc. Natl. Acad. Sci.* *113*, E3548–E3557.
- de Berker, A.O., Rutledge, R.B., Mathys, C., Marshall, L., Cross, G.F., Dolan, R.J., and Bestmann, S. (2016). Computations of uncertainty mediate acute stress responses in humans. *Nat. Commun.* *7*, 10996.
- Botvinick, M.M., Braver, T.S., Barch, D.M., Carter, C.S., and Cohen, J.D. (2001). Conflict monitoring and cognitive control. *Psychol. Rev.* *108*, 624.
- Britten, K.H., Shadlen, M.N., Newsome, W.T., and Movshon, J.A. (1993). Responses of neurons in macaque MT to stochastic motion signals. *Vis. Neurosci.* *10*, 1157–1169.
- Busse, L., Ayaz, A., Dhruv, N.T., Katzner, S., Saleem, A.B., Schölvinck, M.L., Zaharia, A.D., and Carandini, M. (2011). The Detection of Visual Contrast in the Behaving Mouse. *J. Neurosci.* *31*, 11351–11361.
- Ebitz, R.B., and Platt, M.L. (2015). Neuronal Activity in Primate Dorsal Anterior Cingulate Cortex Signals Task Conflict and Predicts Adjustments in Pupil-Linked Arousal. *Neuron* *85*, 628–640.
- Eldar, E., Cohen, J.D., and Niv, Y. (2013). The effects of neural gain on attention and learning. *Nat. Neurosci.* *16*, 1146–1153.
- Fernberger, S.W. (1920). Interdependence of judgments within the series for the method of constant stimuli. *J. Exp. Psychol.* *3*, 126.
- Fetsch, C.R., Kiani, R., and Shadlen, M.N. (2014). Predicting the Accuracy of a Decision: A Neural Mechanism of Confidence. *Cold Spring Harb. Symp. Quant. Biol.* *79*, 185–197.
- Fründ, I., Wichmann, F.A., and Macke, J.H. (2014). Quantifying the effect of intertrial dependence on perceptual decisions. *J. Vis.* *14*, 9–9.
- Gao, J., Wong-Lin, K., Holmes, P., Simen, P., and Cohen, J.D. (2009). Sequential Effects in Two-Choice Reaction Time Tasks: Decomposition and Synthesis of Mechanisms. *Neural Comput.* *21*, 2407–2436.
- de Gee, J.W., Knapen, T., and Donner, T.H. (2014). Decision-related pupil dilation reflects upcoming choice and individual bias. *Proc. Natl. Acad. Sci.* *111*, E618–E625.
- Gold, J.I., and Shadlen, M.N. (2007). The Neural Basis of Decision Making. *Annu. Rev. Neurosci.* *30*, 535–574.
- Hick, W.E. (1952). On the rate of gain of information. *Q. J. Exp. Psychol.* *4*, 11–26.
- Jazayeri, M., and Movshon, J.A. (2006). Optimal representation of sensory information by neural populations. *Nat. Neurosci.* *9*, 690–696.
- Joshi, S., Li, Y., Kalwani, R.M., and Gold, J.I. (2016). Relationships between Pupil Diameter and Neuronal Activity in the Locus Coeruleus, Colliculi, and Cingulate Cortex. *Neuron* *89*, 221–234.
- Kahnt, T., Grueschow, M., Speck, O., and Haynes, J.-D. (2011). Perceptual Learning and Decision-Making in Human Medial Frontal Cortex. *Neuron* *70*, 549–559.
- Kepecs, A., and Mainen, Z.F. (2012). A computational framework for the study of confidence in humans and animals. *Philos. Trans. R. Soc. Lond. B Biol. Sci.* *367*, 1322–1337.
- Kepecs, A., Uchida, N., Zariwala, H.A., and Mainen, Z.F. (2008). Neural correlates, computation and behavioural impact of decision confidence. *Nature* *455*, 227–231.
- Kiani, R., Hanks, T.D., and Shadlen, M.N. (2008). Bounded integration in parietal cortex underlies decisions even when viewing duration is dictated by the environment. *J. Neurosci.* *28*, 3017–3029.
- Kiani, R., Corthell, L., and Shadlen, M.N. (2014). Choice Certainty Is Informed by Both Evidence and Decision Time. *Neuron* *84*, 1329–1342.
- Komura, Y., Nikkuni, A., Hirashima, N., Uetake, T., and Miyamoto, A. (2013). Responses of pulvinar neurons reflect a subject's confidence in visual categorization. *Nat. Neurosci.* *16*, 749–755.
- Lak, A., Costa, G.M., Romberg, E., Koulakov, A.A., Mainen, Z.F., and Kepecs, A. (2014). Orbitofrontal cortex is required for optimal waiting based on decision confidence. *Neuron* *84*, 190–201.
- Law, C.-T., and Gold, J.I. (2008). Neural correlates of perceptual learning in a sensory-motor, but not a sensory, cortical area. *Nat. Neurosci.* *11*, 505–513.
- Law, C.-T., and Gold, J.I. (2009). Reinforcement learning can account for associative and perceptual learning on a visual-decision task. *Nat. Neurosci.* *12*, 655–663.

- Lempert, K.M., Chen, Y.L., and Fleming, S.M. (2015). Relating Pupil Dilation and Metacognitive Confidence during Auditory Decision-Making. *PLOS One* 10, e0126588.
- Luce, R.D. (1991). *Response Times: Their Role in Inferring Elementary Mental Organization* (Oxford University Press).
- Machens, C.K., Romo, R., and Brody, C.D. (2005). Flexible control of mutual inhibition: a neural model of two-interval discrimination. *Science* 307, 1121–1124.
- McDougal, D.H., and Gamlin, P.D.R. (2008). Pupillary Control Pathways. In *The Senses: A Comprehensive Reference*, T.D. Albright, T.D. Albright, R.H. Masland, P. Dallos, D. Oertel, S. Firestein, G.K. Beauchamp, M.C. Bushnell, A.I. Basbaum, J.H. Kaas, et al., eds. (New York: Academic Press), pp. 521–536.
- McGinley, M.J., David, S.V., and McCormick, D.A. (2015). Cortical Membrane Potential Signature of Optimal States for Sensory Signal Detection. *Neuron* 87, 179–192.
- Miller, E.K., and Cohen, J.D. (2001). An Integrative Theory of Prefrontal Cortex Function. *Annu. Rev. Neurosci.* 24, 167–202.
- Murphy, P.R., van Moort, M.L., and Nieuwenhuis, S. (2016). The pupillary orienting response predicts adaptive behavioral adjustment after errors. *PLOS One* 11, e0151763.
- Nassar, M.R., Rumsey, K.M., Wilson, R.C., Parikh, K., Heasly, B., and Gold, J.I. (2012). Rational regulation of learning dynamics by pupil-linked arousal systems. *Nat. Neurosci.* 15, 1040–1046.
- O'Reilly, J.X., Schüffelgen, U., Cuell, S.F., Behrens, T.E.J., Mars, R.B., and Rushworth, M.F.S. (2013). Dissociable effects of surprise and model update in parietal and anterior cingulate cortex. *Proc. Natl. Acad. Sci.* 110, E3660–E3669.
- Pisauro, M.A., Benucci, A., and Carandini, M. (2016). Local and global contributions to hemodynamic activity in mouse cortex. *J. Neurophysiol.* jn.00125.2016.
- Preuschoff, K., 't Hart, B.M., and Einhäuser, W. (2011). Pupil Dilation Signals Surprise: Evidence for Noradrenaline's Role in Decision Making. *Front. Neurosci.* 5, 115.
- Reimer, J., Froudarakis, E., Cadwell, C.R., Yatsenko, D., Denfield, G.H., and Tolias, A.S. (2014). Pupil Fluctuations Track Fast Switching of Cortical States during Quiet Wakefulness. *Neuron* 84, 355–362.
- Sanders, J.I., Hangya, B., and Kepecs, A. (2016). Signatures of a Statistical Computation in the Human Sense of Confidence. *Neuron* 90, 499–506.
- Simoncelli, E.P., and Heeger, D.J. (1998). A model of neuronal responses in visual area MT. *Vision Res.* 38, 743–761.
- Teichert, T., Yu, D., and Ferrera, V.P. (2014). Performance Monitoring in Monkey Frontal Eye Field. *J. Neurosci.* 34, 1657–1671.
- Tervo, D.G.R., Proskurin, M., Manakov, M., Kabra, M., Vollmer, A., Branson, K., and Karpova, A.Y. (2014). Behavioral Variability through Stochastic Choice and Its Gating by Anterior Cingulate Cortex. *Cell* 159, 21–32.
- Varazzani, C., San-Galli, A., Gilardeau, S., and Bouret, S. (2015). Noradrenaline and dopamine neurons in the reward/effort trade-off: a direct electrophysiological comparison in behaving monkeys. *J. Neurosci.* 35, 7866–7877.
- Vinck, M., Batista-Brito, R., Knoblich, U., and Cardin, J.A. (2015). Arousal and Locomotion Make Distinct Contributions to Cortical Activity Patterns and Visual Encoding. *Neuron* 86, 740–754.
- Wang, C.-A., and Munoz, D.P. (2015). A circuit for pupil orienting responses: implications for cognitive modulation of pupil size. *Curr. Opin. Neurobiol.* 33, 134–140.
- Wessel, J.R., Danielmeier, C., and Ullsperger, M. (2011). Error Awareness Revisited: Accumulation of Multimodal Evidence from Central and Autonomic Nervous Systems. *J. Cogn. Neurosci.* 23, 3021–3036.
- Yellin, D., Berkovich-Ohana, A., and Malach, R. (2015). Coupling between pupil fluctuations and resting-state fMRI uncovers a slow build-up of antagonistic responses in the human cortex. *NeuroImage* 106, 414–427.
- Yeung, N., Botvinick, M.M., and Cohen, J.D. (2004). The neural basis of error detection: conflict monitoring and the error-related negativity. *Psychol. Rev.* 111, 931.
- Yu, A.J., and Cohen, J.D. (2008). Sequential effects: Superstition or rational behavior? *Adv. Neural Inf. Process. Syst.* 21, 1873–1880.

Reviewers' Comments:

Reviewer #1 (Remarks to the Author):

Overall this is very well carefully analyzed paper in my opinion.

Reviewer #3 (Remarks to the Author):

The authors have done a commendable job of addressing my concerns in the revised manuscript and I now enthusiastically their submission.

Reviewer #4 (Remarks to the Author):

The authors have done an extensive revision with several new analyses, and a more focused discussion. Overall, the paper is excellent, and all my previous concerns have been resolved.

Dr. Sven Bestmann